# TrimNN: characterizing cellular community motifs for studying multicellular topological organization in complex tissues

Yang Yu [1,9], Shuang Wang[2,9], Jinpu Li [3], Meichen Yu [4], Kyle McCrocklin[5], Jing-Qiong Kang [6], Anjun Ma [7,8], Qin Ma [7,8] ✉, Dong Xu [1,3] ✉ & Juexin Wang [5] ✉

The spatial organization of cells plays a pivotal role in shaping tissue functions and phenotypes in various biological systems and diseased microenvironments. However, the topological principles governing interactions among cell types within spatial patterns remain poorly understood. Here, we present the **tri**angulation cellular community **m**otif **n**eural **n**etwork (TrimNN), a graph-based deep learning framework designed to identify conserved spatial cell organization patterns, termed cellular community (CC) motifs, from spatial transcriptomics and proteomics data. TrimNN employs a semi–divide-and-conquer approach to efficiently detect overrepresented topological motifs of varying sizes in a triangulated space. By uncovering CC motifs, TrimNN reveals key associations between spatially distributed cell-type patterns and diverse phenotypes. These insights provide a foundation for understanding biological and disease mechanisms and offer potential biomarkers for diagnosis and therapeutic interventions.

Various cells work together within spatial arrangements in tissue to support organ homeostasis and function[1]. Deciphering the multicellular organization is key to understanding the relationship between spatial structure and tissue biological and pathological functions[2]. Emerging spatial omics approaches, including spatially resolved transcriptomics[3] and spatial proteomics[4], enable investigation of the mechanisms governing the spatial organization of different cell types in a specific tissue. Within a region of interest (ROI) in spatial omics, cellular neighborhoods (CNs) define local cell type enrichment patterns in cellular communities (CCs), and decoding function-related conservative spatial features in CNs is one of the primary spatial omics data analysis tasks[4].

Most existing data analysis approaches adopt the top-down strategy to describe the cell organizations. This strategy mainly relies on clustering strategies to identify the cell type compositions as common patterns. Deep learning approaches, including SPACE-GM[5], CytoCommunity[6], CellCharter[7], and BANKSY[8], typically learn low-dimensional embeddings of the nodes in corresponding CNs and then apply clustering approaches to these embeddings. However, clustering approaches suffer the following challenges in dissecting and interpreting highly heterogeneous, dynamically evolving cell systems[9]. First, clustering results usually become less stable when samples contain cells under active state transition, which is common in disease

[1]Department of Electrical Engineering and Computer Science, Christopher S. Bond Life Sciences Center, University of Missouri, Columbia, MO, USA. [2]Department of Computer Science, Luddy School of Informatics, Computing, and Engineering, Indiana University Bloomington, Bloomington, IN, USA. [3]Institute for Data Science and Informatics, University of Missouri, Columbia, MO, USA. [4]Indiana Alzheimer's Disease Research Center, Center for Neuroimaging, Department of Radiology and Imaging Sciences, Stark Neurosciences Research Institute, Indiana University School of Medicine, Indianapolis, IN, USA. [5]Department of Biomedical Engineering and Informatics, Luddy School of Informatics, Computing, and Engineering, Indiana University Indianapolis, Indianapolis, IN, USA. [6]Department of Neurology & Pharmacology, Vanderbilt University Medical Center, Vanderbilt Kennedy Center of Human Development, Vanderbilt University, Nashville, TN, USA. [7]Department of Biomedical Informatics, College of Medicine, The Ohio State University, Columbus, OH, USA. [8]Pelotonia Institute for Immuno-Oncology, The James Comprehensive Cancer Center, The Ohio State University, Columbus, OH, USA. [9]These authors contributed equally: Yang Yu, Shuang Wang. ✉e-mail: qin.ma@osumc.edu; xudong@missouri.edu; wangjuex@iu.edu

or developmental processes[10]. Second, clusters identified by these top-down approaches are often described as percentages of cell-type compositions. These clustering presentations lack formulations in topologically representing the geometrical cell-type interactions or are difficult to interpret biologically. Last, these top-down results essentially depend on the presence of batch effects, where CNs separate primarily by samples as technical covariates rather than biological features[3]. These batch effects make it easy to overfit the models but difficult to validate across different datasets[5].

Considering the preceding limitations of top-down strategies, we instead use a bottom-up strategy to identify CC motifs as recurring significant interconnections between cells. In the spatial omics–derived CC, we hypothesize that CC motifs can be represented as topological building blocks of multicellular organization consistent across different samples and associated with key biological processes and functions. CC motifs are biologically interpretable spatial patterns of the combined cell types, which provide topological information beyond clusters and explicitly link to the biological and pathological mechanisms through distinct cell–cell communications, highly expressed genes and pathways[11]. This concept is related to the functional tissue units (FTUs)[12], but CC motifs are even smaller in the scale of cell locations and cell types, which provides more details for understanding and modeling the healthy physiological function of the organ and functional-related changes during disease states. Currently, size 1–3 motif analysis[4] makes up most of the spatial omics studies, where size-1 motifs are single nodes that can be treated as cell-type compositions, size-2 motifs are double nodes linked by edges, and size-3 motifs are triple nodes within triangles. Nevertheless, biologists have found that sizable CC motifs with more nodes than triangles substantially correlate with patient survival and phenotypical features in colorectal cancer (CRC)[13], kidney diseases[14], maternal–fetal interface[15], and many other biological contexts.

In practice, identifying the most overrepresented CC motifs composing multicellular organization is still computationally expensive with (*i*) subgraph matching[16], which counts the occurrence of a given motif on the query graph, and (*ii*) pattern growth[17], which finds the motifs with the most significant occurrence. It is known that subgraph matching is NP-complete[16], which makes the node type combination alone super-exponential. Existing approaches include permutation[11], edge sampling (e.g., mfinder[18]), node sampling (e.g., FANMOD[19]), and global pruning (e.g., Ullmann[20] and VF2[21]). A computationally feasible approach is still lacking to analytically identify conservative, interpretable, and generalizable spatial rules of cellular organization in different sizes across different samples of spatial omics.

Here, we propose the triangulation cellular community motif neural network (TrimNN), a graph-based deep learning approach to analyze spatial transcriptomics and proteomics data using a bottom-up strategy (Supplementary Fig. 1). Within the input spatial omics samples, CC is defined based on the cells as nodes, the node types represent different cell types, and the edges encode physical proximity inferred unidirectional as the spatial cell-cell relation from Delaunay triangulation[22] based on nodes coordinates from ROI. TrimNN estimates overrepresented size-*K* CC motifs in the CC of spatial omics using graph isomorphism networks[23] (GIN) empowered by positional encoding[24] (PE). In various spatial transcriptomics and spatial proteomics case studies, TrimNN identifies computationally significant and biologically meaningful CC motifs to differentiate patient survival in CRC studies and represents pathologically related cell type organization in neurodegenerative diseases and colorectal carcinoma studies. Notably, the identified sizable CC motifs demonstrate their potential as interpretable topological prognostic biomarkers linking the topological structural organization of cell types at microscopic levels to phenotypes at macroscopic levels, which cannot be inferred

by other existing tools. The source code of TrimNN is publicly available at https://github.com/yuyang-0825/TrimNN.

## Results

### TrimNN quantifies multicellular organization with sizable CC motifs

A schematic diagram of the proposed TrimNN and its analytic workflow is shown in Fig. 1A. These identified CC motifs are biologically interpretable through a set of downstream analyses, including motif visualization, cellular-level interpretation within cell–cell communication analysis, gene-level interpretation within differentially expressed gene and pathway analysis, and phenotypical analysis within the availability of phenotypical information (Fig. 1B). TrimNN is constructed on an empowered GIN to estimate the occurrence of the query on the target graph. On the CC as a triangulated graph built from spatial omics, TrimNN builds a supervised graph learning model by simplifying the graph constraints and incorporating the inductive bias within triangles derived from Delaunay triangulation. TrimNN decomposes the regression task in occurrence counting of the query graphs into many trackable binary classification tasks modeled by the sub-TrimNN module. Inspired by the idea of NSIC[25], this method is trained on representative pairs of the predefined query subgraphs and the target triangulated cell graphs as a binary classification task. This graph representation framework builds upon GIN and adopts a shortest distance–based PE[24], modeling the symmetric space to increase the expressive power. Additionally, TrimNN adopts a semi–divide-and-conquer strategy to estimate the abundance of the query by summarizing the enumeration of single classification tasks by a sub-TrimNN module on each node's enclosed graph. Given the size of the query subgraph, our framework uses an enumeration approach to estimate the most overrepresented CC motifs with possible cell types and topology. Then, we search to infer CC motifs in different sizes incrementally. The details of the architecture of TrimNN are shown in Supplementary Fig. 2.

We hypothesize CC motifs as the countable recurring spatial patterns of various cell types are robust within noises to represent and quantify multicellular organization. We performed simulations to mimic different levels of noises, including cell missing from the cell capture imperfection of sequencing technology (Fig. 2A), cell coordinate shifting from technological errors (Fig. 2B), and cell type misclassification from annotation errors in data analytics (Fig. 2C). We noticed diverse noises do not influence the relative ranking of CC motif abundance, which remained robustly consistent in most scenarios (Fig. 2A and Supplementary Fig. 3). Even under extreme cases with a noise ratio of 0.4 and 0.5, the Spearman correlation between abundance rankings before and after poised noise remained relatively stable in cell dropout and cell coordinate perturbations. When the noise level is high, the correlation values deteriorate in large motif sizes for cell type misclassification, which is unlikely to occur in practical scenarios.

### TrimNN accurately identifies overrepresented CC motifs in Cellular Neighborhoods

On a modified subgraph matching task as a binary classification of motif existence in a triangulated graph, TrimNN outperformed the competitive methods in all scenarios in most criteria in synthetic spatial omics data, including VF2, the original regression-based neural network method NSIC[25], and TrimNN-RGIN with the proposed formulation but using NSIC's RGIN network architecture. Especially on large-size CC motifs, TrimNN demonstrated significant performance improvements with TrimNN-RGIN, highlighting its architecture's capacity (Fig. 2D and Supplementary Data 1).

TrimNN accurately identified the top overrepresented CC motifs. On a pattern growth challenge to determine the ranking of CC motif abundance, TrimNN outperformed competitive methods consistently

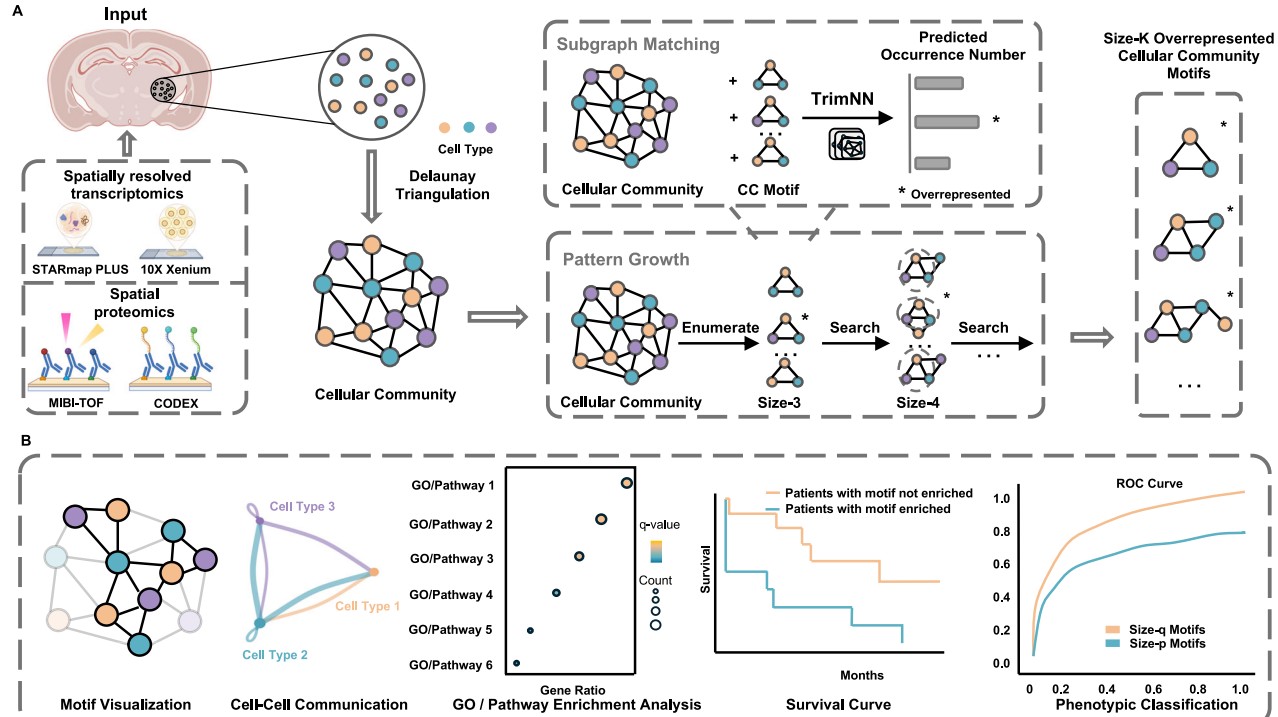

**Fig. 1 | TrimNN analysis workflow. A** Spatially resolved transcriptomics (e.g., STARmap PLUS and 10X Xenium) and spatial proteomics data (e.g., MIBI-TOF and CODEX) are used as input to generate corresponding CCs with spatial coordinates and Delaunay triangulation. TrimNN is trained on representative pairs of query motifs and target triangulated graphs at scale. Given a specific query, TrimNN identifies its occurrence in the target CC in the subgraph matching process by decomposing this regression task into many binary classification problems, where each classification predicts whether the query exists in the target graph as the enclosed graph of each node. Enumerating possible motifs at size-$K$, TrimNN identifies the most overrepresented motifs. Then, the pattern growth process adopts a heuristic search for their successor size-$k$+1 motifs. Here, we take size-3 CC motifs as an example. After subgraph matching and pattern growth, TrimNN estimates overrepresented CC motifs. Created in BioRender. Yu, Y. (2025) BioRender.com/mfm4ta4. **B** These CC motifs can be biologically interpreted in the downstream analysis, including visualization, cellular-level interpretation within cell−cell communication analysis, gene-level interpretation within differentially expressed gene analysis (e.g., GO enrichment analysis and pathway enrichment analysis), and phenotypical analysis within the availability of phenotypical information (e.g., survival curve and phenotypic classification analysis). CC: cellular community.

in different sizes and cell types in synthetic spatial omics data. Both TrimNN and TrimNN-RGIN outperformed NSIC by a large margin in most scenarios and criteria, which highlights the capability of the proposed problem formulation. Notably, TrimNN demonstrated an average improvement over NSIC by approximately 20 to 60 times in root mean square error (RMSE) (Fig. 2E and Supplementary Data 2). Besides the criteria in absolute occurrence value, the relative value of the ranking index also supported TrimNN's capacity in Supplementary Fig. 4A and Supplementary Data 3.

TrimNN is highly scalable in identifying large-size CC motifs. Because scalability plays a vital role in the study, we compared the computational time on target-triangulated graphs with varying node sizes. We observed that TrimNN, TrimNN-RGIN, and NSIC exhibit linear scalability with increasing node sizes, while TrimNN continuously consumed less computational time (Fig. 2F). TrimNN was especially more efficient than TrimNN-RGIN with a simpler network architecture using the same problem setting. In contrast, the classical enumeration-based VF2 method grew exponentially, where its runtime made it unacceptable in most scenarios. In practical usage, on typical spatial omics data with thousands of cells of dozens of cell types, TrimNN robustly infers large-size CC motifs accurately in seconds, which is unattainable through conventional methods.

Together with GIN, PE increases the expressive power of TrimNN. In challenging tasks with larger-sized motifs, ablation tests showed that integrating PE improved GNN (Graph Neural Network) performance compared with TrimNN-RGIN without PE and a complex GRU module (Fig. 2G and Supplementary Data 4). In addition, GIN, as the critical component in TrimNN, was effective by replacing it with other

graph neural network models, including Graph Convolutional Networks and Graph Transformer[26], keeping other components and parameters constant (Supplementary Fig. 4B and Supplementary Data 5). This result aligned with theoretical analyses that GIN is a powerful 1-order graph neural network[23]. Meanwhile, it was shown that TrimNN requires sufficient training data to learn the complex relationships (Supplementary Fig. 4C and Supplementary Data 6).

## TrimNN identifies representative CC motifs that accurately differentiate the severity of colorectal cancer patients

In addition to the above simulation studies, we showed that the CC motifs inferred by TrimNN are intrinsic representations to differentiate phenotypes of the CC. In a proteomics study comprising 17 low-risk (Crohn's-like lymphoid reaction, CLR) and 18 high-risk (diffuse inflammatory infiltration, DII) patients, using Co-Detection by Indexing (CODEX)[13] on CRC, we performed a CC motif analysis using TrimNN on 140 tissue regions and identified the most abundant CC motifs in size-1 to size-4. Traditional machine learning approaches, such as logistic regression (LR), were adopted using relative ranking indices to quantify motif occurrence as features. Because the original publication annotated 29 cell types, we chose 29 as the fixed number of features in supervised learning to classify CLR and DII. Within tenfold cross-validation following the same protocol as CytoCommunity[6], the ROC-AUC results of LR were 0.77, 0.76, 0.79, and 0.76 (Fig. 3A) for size-1 to size-4 CC motifs, respectively. This LR model with 29 CC motif features outperformed CytoCommunity's extensive GNN computation performance using a default of 512 dimensions of embeddings as features (ROC-AUC: 0.71). Notably, if the feature number increased to the top

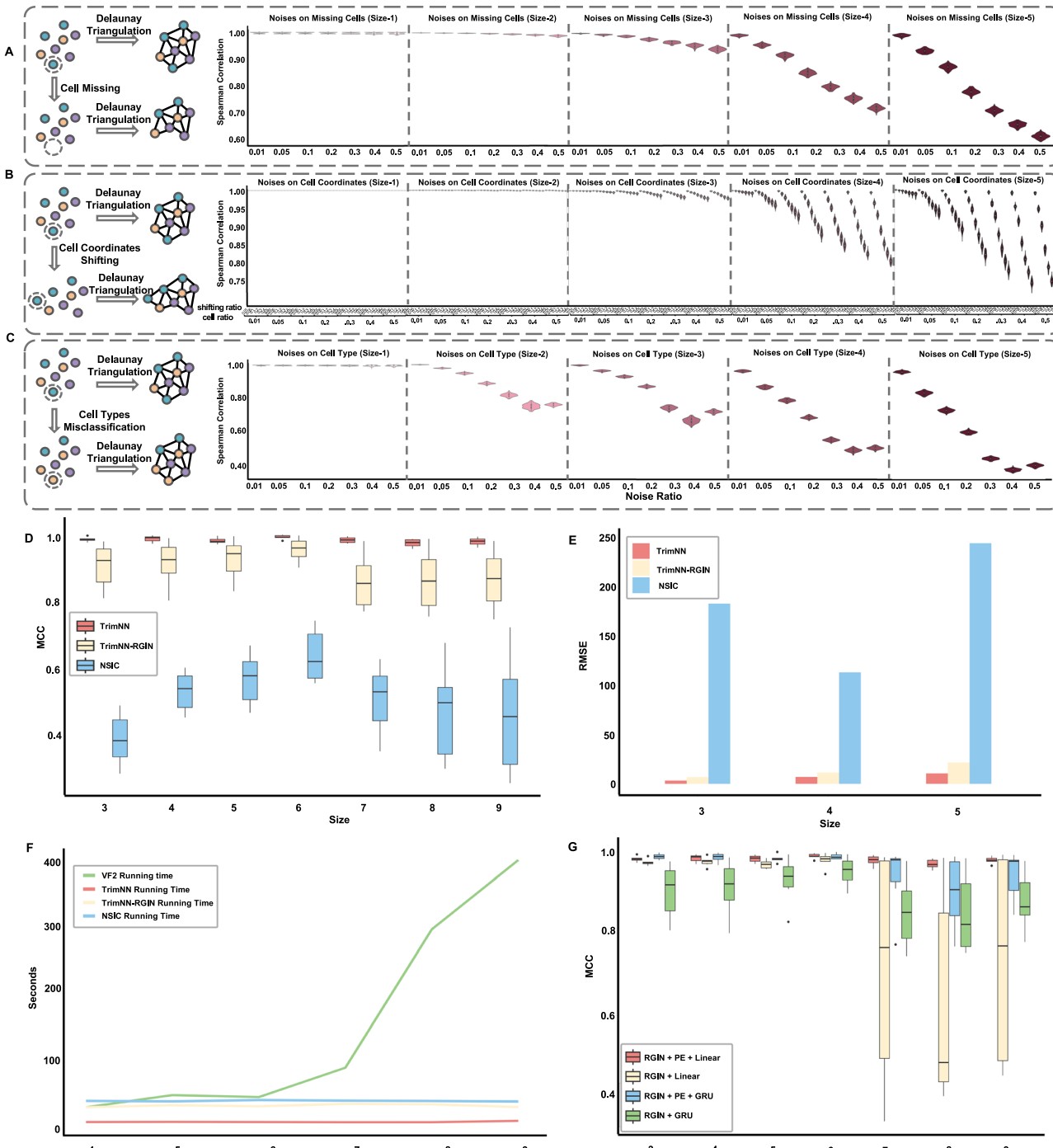

**Fig. 2 | The performance of TrimNN on spatial omics. A** Simulations of missing cell effects on CC motifs, represented as the Spearman correlation between abundance rankings of all the possible motifs before and after simulated noises at cell proportions of 0.01, 0.05, 0.1, 0.2, 0.3, 0.4, and 0.5 within CC motifs in size-1, size-2, size-3, size-4, and size-5 ($n = 100$). **B** Simulations of cell coordinate shifting effects on CC motifs, represented as the Spearman correlation between abundance rankings of all the possible motifs before and after simulated noises with different levels of noises of 0.01, 0.05, 0.1, 0.2, 0.3, 0.4, and 0.5 at cell proportions of 0.01, 0.05, 0.1, 0.2, 0.3, 0.4, and 0.5 within CC motifs in size-1, size-2, size-3, size-4, and size-5 ($n = 100$). **C** Simulations of cell-type misclassification effects on CC motifs, represented as the Spearman correlation between abundance rankings of all the possible motifs before and after simulated noises at cell proportions of 0.01, 0.05, 0.1, 0.2, 0.3, 0.4, and 0.5 within CC motifs in size-1, size-2, size-3, size-4, and size-5 ($n = 100$). **D** Benchmarking the performance of TrimNN, TrimNN-RGIN, and NSIC on

independent simulated data for subgraph matching ($n = 3000$). The *X*-axis represents different sizes of CC motifs, and the *Y*-axis indicates the MCC (Matthews Correlation Coefficient) values. **E** Performance comparison of TrimNN, TrimNN-RGIN, and NSIC in identifying occurrences of CC motifs in diverse simulated datasets. The *Y*-axis is the RMSE (Root Mean Square Error) value. **F** Scalability of TrimNN. The *X*-axis represents the size of the triangulated graph, and the *Y*-axis indicates the runtime on a workstation equipped with an Intel Xeon Gold 6338 CPU and 80 G RAM. **G** Ablation tests on performance comparison adding the positional encoding of TrimNN model ($n = 3000$). The *X*-axis represents different sizes of CC motifs, and the *Y*-axis indicates the MCC values. CC: cellular community. On each box, the central mark indicates the median, and the bottom and top edges of the box indicate the 25th and 75th percentiles. The whiskers extend to the most extreme data points without outliers, and the outliers are plotted individually as circles. Source data are provided as a Source Data file.

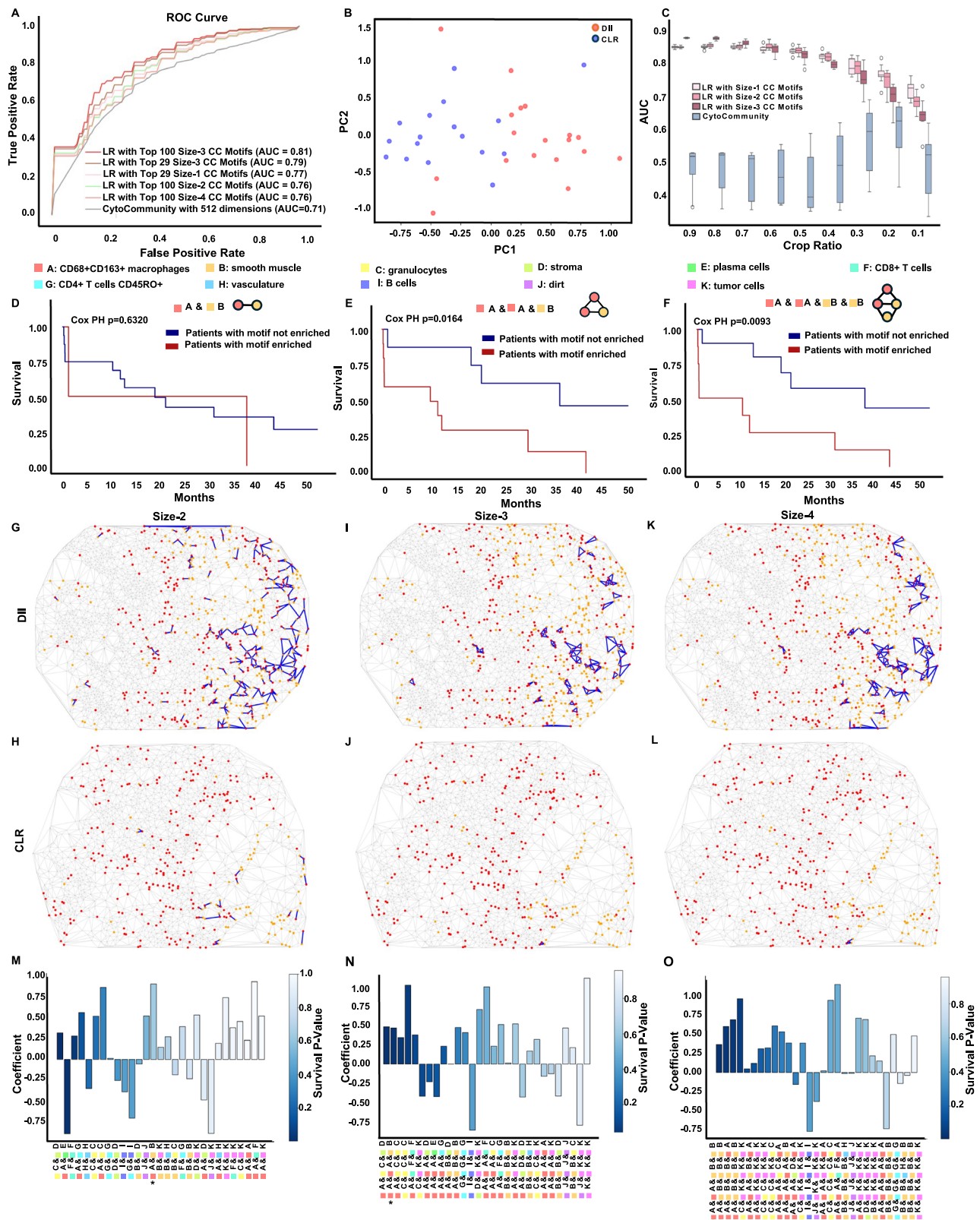

100, the LR model on size-3 motifs achieved an ROC-AUC of 0.81. To investigate the robustness of the model against potential overfitting, additional experiments were performed using 10 times 10-fold cross-validation on the LR model with the top 5, 10, 15, and 20 size-3 CC motif features, as well as CytoCommunity using reduced 29-dimensional embedding (Supplementary Fig. 5 and Supplementary Data 7). The performance of the LR model with diverse small numbers of CC motif

features is relatively stable, suggesting it encounters limited influences from overfitting with LR. In addition, other classical machine learning models, such as Random Forest and Support Vector Machine, were applied to the same classification tasks with the same settings. These models performed similarly to LR, further supporting the representational power of CC motifs (Supplementary Data 8-10). The evaluation

**Fig. 3 | TrimNN analysis in a colorectal cancer study using CODEX. A** The ROC curves of the LR model classify CLR and DII patients using the top CC motifs of size-1 to size-4 as features, and the competitive method CytoCommunity uses learned dimension. The LR model uses features as motif counts from TrimNN and scales between 0 and 1. **B** Visualization of all the samples using the top two principal components from the 29 top size-2 CC motifs. Blue spots denote the CLR patient group and red spots denote the DII patient group. **C** Generalizability of the trained model testing on random cropping of ROI in the samples (*n* = 100). The *X*-axis is the ratio of width and height of the original ROI, and the *Y*-axis is ROC-AUC. On each box, the central mark indicates the median, and the bottom and top edges of the box indicate the 25th and 75th percentiles. The whiskers extend to the most extreme data points without outliers, and the outliers are plotted individually as circles. Survival curves of DII patients with and without enriched motifs, including

**D** size-2 "A & B". Here, cell type CD68+CD163+ macrophages are denoted as "A" and smooth muscles are denoted as "B". **E** size-3 "A & A & B" and **F** size-4 "A & A & B & B". The visualization of spatial localization of size-2 CC motif "A & B" on the CC in **G** patient 3 (DII) on spot 5A and **H** patient 8 (CLR) on spot 16 A. The visualization of the spatial locations of the size-3 motif "A & A & B" in **I** DII spot and **J** CLR spot (same spots as **G** and **H**). The visualization of spatial localization of the size-4 motif "A & A & B & B" in **K** DII spot and **L** CLR spot (same spots as **G** and **H**). All motifs are marked as blue, nodes of cell type "A" are red, and nodes of cell type "B" are orange. The plot of LR coefficients ranked by Cox PH *p*-value of the top 29 CC motifs in **M** size-2, **N** size-3, and **O** size-4. The extent of the blue color represents the Cox PH *p*-value. The *p*-values were derived from the two-sided Wald test. * marks the highlighted motif. LR: logistic regression, CC: cellular community. Source data are provided as a Source Data file.

---

of the comprehensive performance comparison to CytoCommunity and SPACE-GM is provided in Supplementary Data 11.

Besides supervised learning, these CC motifs seemed to capture some intrinsic characteristics in CNs, where CLR and DII demonstrated good visual separation using the top two principal components inferred from the top 29 size-2 motifs (Fig. 3B). Further unsupervised hierarchical clustering showed the different distributions of the top 29 motif abundances among the CLR and DII groups in Supplementary Fig. 6A and 6B (size-2), Supplementary Fig. 6C and 6D (size-3), and Supplementary Fig. 6E and 6F (size-4).

CC motifs in simpler models and fewer numbers of features showed better generalizability across multiple samples in machine learning. When only parts of the CC were available by random cropping the samples, CC motif–based LR methods were very robust in generalizability compared to competitive methods (Fig. 3C). The same trends were observed in distorted samples with simulated noises in cell missing, cell coordinate shifting, and cell type misclassification (Supplementary Fig. 7).

In addition, the enrichment of sizable CC motifs can be used to differentiate patient survival. We identified several size-2, size-3, and size-4 CC motifs that significantly differentiate survival (Cox PH *p* < 0.05) between enriched and non-enriched DII patients, while cell type composition (size-1 motifs) may not necessarily succeed (Supplementary Data 12). With cell type "CD68+CD163+ macrophages" (denoted by "A") and cell type "smooth muscle" (denoted by "B"), the survival curves showed that size-2 motif "A & B" may not have adequately separated survival in the DII patient group (Cox PH *p* = 0.63, shown in Fig. 3D), but including more adjacent nodes with the same cell types, patients with enrichment of size-3 ("A & A & B") and size-4 ("A & A & B & B") CC motifs showed significant lower survival rates (Cox PH *p* = 0.016, shown in Fig. 3E, and Cox PH *p* = 0.0093, shown in Fig. 3F, respectively). To validate the results, we performed additional survival analyses in both COAD and READ cohorts of The Cancer Genome Atlas (TCGA) associated with CD68+CD163+ macrophage marker genes: *CD68, CD163, CD14,* and *ITGAM*[27], and smooth muscle marker genes: *ACTA2, MYH11,* and *MYL9*[28]. We observed no significant differences in survival between patients associated with either cell type (Cox PH *p* > 0.05, Supplementary Figs. 8 and 9). This independent analysis showed consistent results, indicating that cell type compositions, such as size-1 CC motifs, have limited effectiveness in differentiating patient survival in CRC. In addition, the occurrence numbers of these CC motifs among DII and CLR patients were 14,415 and 7004 (ratio 2.06) for size-2 "A & B"; 4176 and 1548 (ratio 2.70) for size-3 "A & A & B"; and 6946 and 2276 (ratio 3.05) for size-4 "A & A & B & B," respectively in each case. All were inferred as significant through the Benjamini-Hochberg adjusted Fisher's exact test. The different distribution of these CC motifs on CCs among DII and CLR spots can be visualized in Fig. 3G and H (size-2), Fig. 3I and J (size-3), and Fig. 3K, L (size-4).

Notably, spatial topology plays a crucial role in linking phenotypes and survival. There were two types of size-3 motifs with cell types

CD68+CD163+macrophages ("A") and smooth muscle ("B"). Compared with "A & A & B", the alternative motif "A & B & B" occurred 4602 and 2255 times among DII and CLR patients, respectively, with a lower ratio of 2.04, and it cannot differentiate survival well (Cox PH *p* = 0.2975). Apparently, these topological differences among the spatial localization of cells in different cell types played different roles biologically and pathologically, where conventional top-down approaches with cell type composition failed to distinguish (Supplementary Fig. 10 and Supplementary Data 13–16).

Furthermore, an LR model provides intrinsic interpretability when differentiating phenotypes. The coefficients of each feature from the LR model demonstrated the importance of CC motifs quantitatively, making the model interpretable (Fig. 3M–O). Notably, all macrophage-related, muscle-related, and significant Cox PH *p*-value motifs in different sizes tended to have high absolute coefficient values. The same interpretable results can also be cross-validated by Shapley value[29] in Supplementary Fig. 6G–I, showing that these macrophage-related and muscle-related CC motifs were essential to differentiating DII patients from CLR patients. Biologically, it was evidenced that macrophages facilitate pancreatic cancer to induce muscle wasting via promoting TWEAK (TNF-like weak inducer of apoptosis) secretion from the tumor[30]. After carefully checking these top motifs in different sizes, we also identified biologically meaningful tumor cells and B cells, which were known to be related to the severity of CRC[31]. Representative tumor and B cell enrichments in DII and CLR samples are shown in Supplementary Fig. 6J, 6K, 6L, and 6M. Our analysis validated the crosstalk between macrophages, muscle wasting, and cancer cachexia through an independent spatial omics study, and TrimNN identified CC motifs in a data-driven approach as robust interpretable representations in CNs.

## TrimNN identifies CC motifs revealing diverse roles in Alzheimer's disease using spatial transcriptomics data

Next, we showed TrimNN's capability to identify diverse spatially distributed CC motifs corresponding to multiple biological and pathological mechanisms in complex diseases. It is known that the interaction between CTX (Cortex) excitatory neurons and Microglia is significantly disrupted by neuroinflammation in Alzheimer's disease (AD)[32]. However, their topological combinations, particularly their relationship with amyloid-β on the cellular level, are still unknown[33]. In a study, we performed TrimNN on an AD mouse brain with 8-month-old and 13-month-old samples sequenced by STARmap PLUS spatially resolved transcriptomics[34]. There were two replicates for both disease and control conditions at each time point. The transcriptomics data included 2,766 genes and two proteomics channels representing AD markers of amyloid-β and tau pathologies at subcellular resolution.

On the derived CCs, size-3 triangle-like CC motifs composed of cell types CTX excitatory neurons and Microglia were identified significant between AD (Fig. 4A) and control (Fig. 4B). These significant CC motifs included CTX excitatory neurons-CTX excitatory neurons-CTX excitatory neurons (CCC), CTX excitatory neurons-CTX

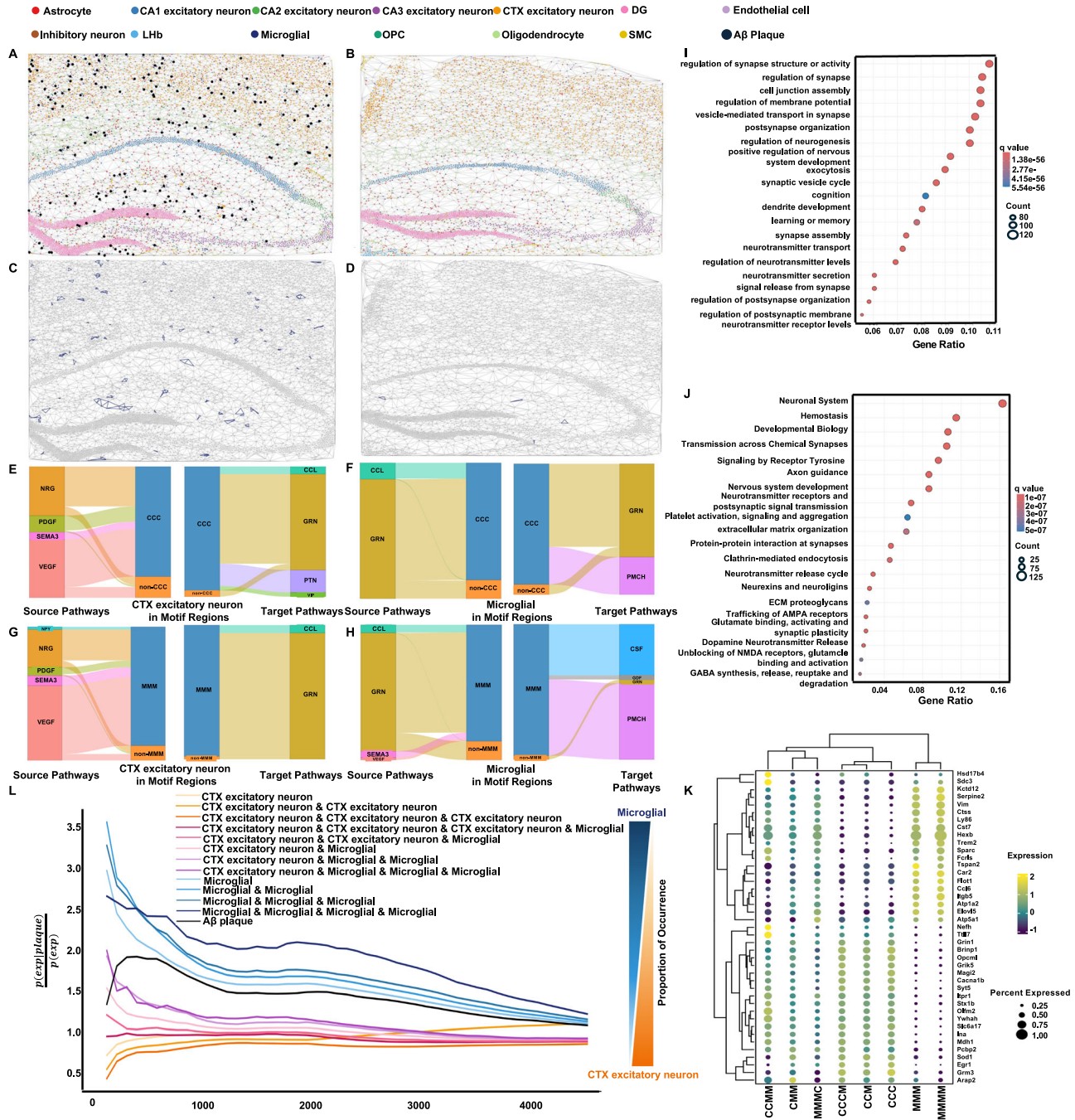

**Fig. 4 | TrimNN analysis in an AD mouse study sequenced by STARmap PLUS.** CCs of 13-month-old of **A**. AD and **B** control sample replicate 1 are obtained using Delaunay triangulation, where black spots are amyloid-β in the AD sample. The spatial locations of the identified motif with all Microglia cells ("MMM" motif, where "M" denotes cell type Microglia) in 13-month-old replicate 1 of **C**. AD and **D** control mouse samples. "MMM" motifs are marked as purple. Cell–cell communication analysis demonstrates the ligand–receptor differences between motif regions and non-motif regions as river plots, including **E** cell type CTX excitatory neurons (denoted as "C") as source (left) and target (right), **F** cell type Microglia as source (left) and target (right) in regions with and without "CCC" motif. Similarly, **G** and **H** are cell types of CTX excitatory neurons and Microglia as source and target in

regions with and without the "MMM" motif. **I** GO enrichment analysis of Biological Processes and **J** Pathway enrichment analysis on DEGs between regions containing and not containing the "MMM" motif in 13-month-old AD samples. **K** Expression of marker genes for cell types "C" and "M" related size-3 and size-4 motifs. **L** Spatial co-occurrence of different CC motifs with respect to amyloid-β as computed using Squidpy. Microglia-related motifs have an even higher spatial co-occurrence probability compared to the amyloid-β plaque, and CTX excitatory neurons–related motifs have lower spatial co-occurrence probabilities compared to the amyloid-β plaque. CC: cellular community. Source data are provided as a Source Data file.

excitatory neurons-Microglia (CCM), CTX excitatory neurons-Microglia-Microglia (CMM), and Microglia-Microglia-Microglia (MMM) (Benjamini–Hochberg adjusted Fisher's exact test in 8-month-old replicate 1 with $p = 6.12e−32$, $p = 3.23e−20$, $p = 4.30e−34$, and $p = 9.74e−07$, respectively). Visualization of an exemplary CC

motif "MMM" demonstrated uneven spatial distribution that differed in AD (Fig. 4C) and control (Fig. 4D). Supplementary Fig. 11 shows the spatial occurrence distribution of the other three motifs. The CCs inferred from all samples are shown in Supplementary Fig. 12 and Supplementary Data 17–28.

Here, we defined motif-enriched regions as expanded regions within three hops of CC motifs in the CC. From the perspective of cell–cell communications, unique ligand–receptor signaling pathway patterns between "CCC" (Fig. 4E, F) and "MMM" (Fig. 4G and H) were identified by motif-enriched and complementary regions on 13-month-old samples using CellChat[35]. Specific to cell type Microglia, motifs "CCC" and "MMM" had dominant ligand–receptor pairs GRN[21] and PMCH to distinguish motif regions from the complementary regions. AD-related ligand–receptors, including GRN, VEGF, PDGF, CCL, VIP, NRG, and SEMA3, were significantly enriched in CC motifs associated with CTX excitatory neurons or Microglia (Supplementary Data 29–32). All the cell–cell communication results on CC motifs are detailed in Supplementary Figs. 13–21 and Supplementary Data 33–40. These differences in cell–cell communications were independently validated by DeepTalk[36], incorporating long-range cellular interactions. Particularly, CSF and VEGF pathways exhibited significant differences between "CCC" and " non- CCC" regions in Microglia-to-CTX excitatory neurons cell–cell communication (Mann–Whitney $U$ test, $p = 0.033$, and $p = 0.003$, Supplementary Figs. 22–25). Considering cell–cell communication among all the cell types, all ligand–receptor pairs identified by CellChat show significant differences with and outside the identified motif regions (Mann–Whitney $U$ test, $p$-value < 0.001, Supplementary Figs. 26 and 27).

Then we analyzed the gene-level characteristics of identified CC motifs. Comparing "MMM" motif-enriched and complementary regions, differentially expressed genes (DEGs) were identified as significant ($p$-value < 0.05) using the Wilcoxon rank-sum test, including *Plekha1*, *Ctsb*, and *Sort1* in 8-month-old samples (Supplementary Data 44), and *App*, *Plekha1*, *Clu*, *Ptk2b*, *Sort1*, *Bin1*, and *Ctsb* in 13-month-old samples (Supplementary Data 48). On DEGs in 13-month-old samples, Gene Ontology (GO) enrichment analysis showed significant vesicle-mediated transport in synapse ($q$-value = 1.68E−107), regulation of synapse structure or activity ($q$-value = 6.45e−106), learning or memory ($q$-value = 1.20e−56), and cognition ($q$-value = 5.54e−56) (Fig. 4I). Neural systems ($q$-value = 3.63e−51), transmission across chemical synapses ($q$-value = 2.05e−33), neurotransmitter receptors and postsynaptic signal transmission ($q$-value = 8.68e−20), and nervous system development ($q$-value = 1.33e−15) were enriched with pathway enrichment analysis (Fig. 4J). For all the detailed results of CC motifs "CCC", "CCM", "CMM", and "MMM", a similar analysis was performed for DEGs (Supplementary Data 41–48), including both GO and pathway enrichment analyses (Supplementary Figs. 28 and 29).

To validate their relations with AD, we compared these motif-related DEGs with 77 AD-associated genes identified from large-scale GWAS analysis[37]. On CC motif "CCC", the *Trem2* gene was exclusively observed in the replicates of the 13-month-old but not 8-month-old AD mouse model, consistent with its role in the late-onset of AD[38]. Similarly, for the motif "MMM", the *Clu* gene was highlighted only in the 13-month-old mouse model, aligning with its direct involvement in the formation process of amyloid-β[39] (Supplementary Fig. 30).

Based on the identified size-3 motifs, we performed pattern growth to identify size-4 motifs using TrimNN (Supplementary Notes 1). Among all the size-4 "CCC" expanded motifs, "CCCM" showed the most significant difference between AD and control samples, while "MMMM" was the most significant size-4 motif expanded from "MMM". Similar to the analysis on size-3 motifs, checking significantly enriched ligand–receptors (Supplementary Data 49–51) and DEGs (Supplementary Data 52–57), these size-4 motifs were related to AD in cell–cell communication analysis (Supplementary Data 58–63), GO enrichment analysis (Supplementary Fig. 31), and pathway enrichment analysis (Supplementary Fig. 32).

Further investigation on DEGs showed diverse groups of CC motifs with expressed markers (Fig. 4K). Size-3 "MMM" and size-4 "MMMM" motifs with homogeneous Microglia had divergent expression patterns. For example, *Hexb* had a higher average expression than the other CC motifs. *Hexb* is known to induce toxic and progressive neuronal damage, which may relate to neurodegenerative dementia[40].

In addition to examining the diversity of CC motifs at the gene level, we investigated whether the identified CC motifs were spatially co-localized with amyloid-β by computing their co-occurrence probabilities using Squidpy[41]. The results showed that Microglia-related CC motifs had an even higher co-occurrence probability with amyloid-β than the spatial expectation, distinguishing them from other CC motifs associated with CTX excitatory neurons (Fig. 4L). Interestingly, the extent of homogeneity of Microglia regions seemed to correspond to a larger co-occurrence probability of amyloid-β. In contrast, the extent of homogeneity of CTX excitatory neurons tallied to a lower co-occurrence probability. This trend prevailed across the whole spectrum of CC motifs composed of Microglia and CTX excitatory neurons in multiple sizes, from a very high ratio of size-4 "MMMM" to a very low ratio of size-3 "CCC".

Differences in both DEGs and spatial co-occurrence suggest the presence of two distinct types of CC motifs related to amyloid-β in AD. One type of CC motif (i.e., "CCC", "CCM", "CMM", and "CCCM") was reluctant to co-localize with amyloid-β. Another kind of CC motif (i.e., "MMM" and "MMMM") was closely co-localized with amyloid-β. These results were consistent with the observation that Microglia, as key mediators in the brain, activate inflammation in the vicinity of amyloid-β deposits, which are directly toxic to the adjacent neurons[42]. Activated Microglia release pro-inflammatory cytokines, such as tumor necrosis factor-alpha (TNF-α) and interleukin-1 beta (IL-1β), can damage excitatory neurons or alter their function[43].

In this case study, TrimNN confirmed known knowledge of AD-related cell types and provided some new insights into spatial biology. As an unbiased data-driven approach, TrimNN independently identified pathologically related spatial characteristics of Microglia and CTX excitatory neurons, along with their topological relations with diverse cell types, as two distinguished CC motifs differ in levels of cell type, gene, and cell–cell communications. Analysis enabled by CC motifs demonstrates an unprecedented spectrum of the spatial relationships between the homogeneity of CTX excitatory neurons/Microglia cell types and the location of amyloid-β. TrimNN accurately captured these spatial co-localization patterns with amyloid-β deposits, providing insights into the onset of AD as the result of interactions between multiple cell types[44], which clustering-based tools may overlook (Supplementary Figs. 33 and 34).

## TrimNN identifies cell type–specific spatial tendencies in a colorectal carcinoma study on spatial proteomics data

Besides the AD study, we also performed TrimNN analysis to explore cell type–specific spatial tendencies in one colorectal carcinoma study. It is known that the tumor microenvironment can significantly influence the interactions between T-cells and epithelial cells through antigen presentation, T-cell activation, and modulation of the tumor microenvironment. However, it is still unknown how the spatial arrangement of these cells is related to effective immune surveillance and the potential for therapeutic interventions[45]. The adopted colorectal carcinoma study investigated 40 ROIs in two colorectal cancer patients and 18 ROIs in two healthy controls using spatial proteomics of multiplexed ion beam imaging using time of flight (MIBI-TOF)[46].

After a comprehensive analysis of size-3 and their related size-4 CC motifs with TrimNN, we defined two types of CC motifs: Shifted Interaction Motifs and Homeostatic Interaction Motifs (Fig. 5A). Shifted Interaction Motifs demonstrated a shift of CC motif abundance from control-enriched (more occurrence in control than disease samples) to disease-enriched (more occurrence in disease than control samples) when expanding from size-3 to size-4. The exemplary size-3 motif "ABC" (Fig. 5B, C) suggested disease progression when involving

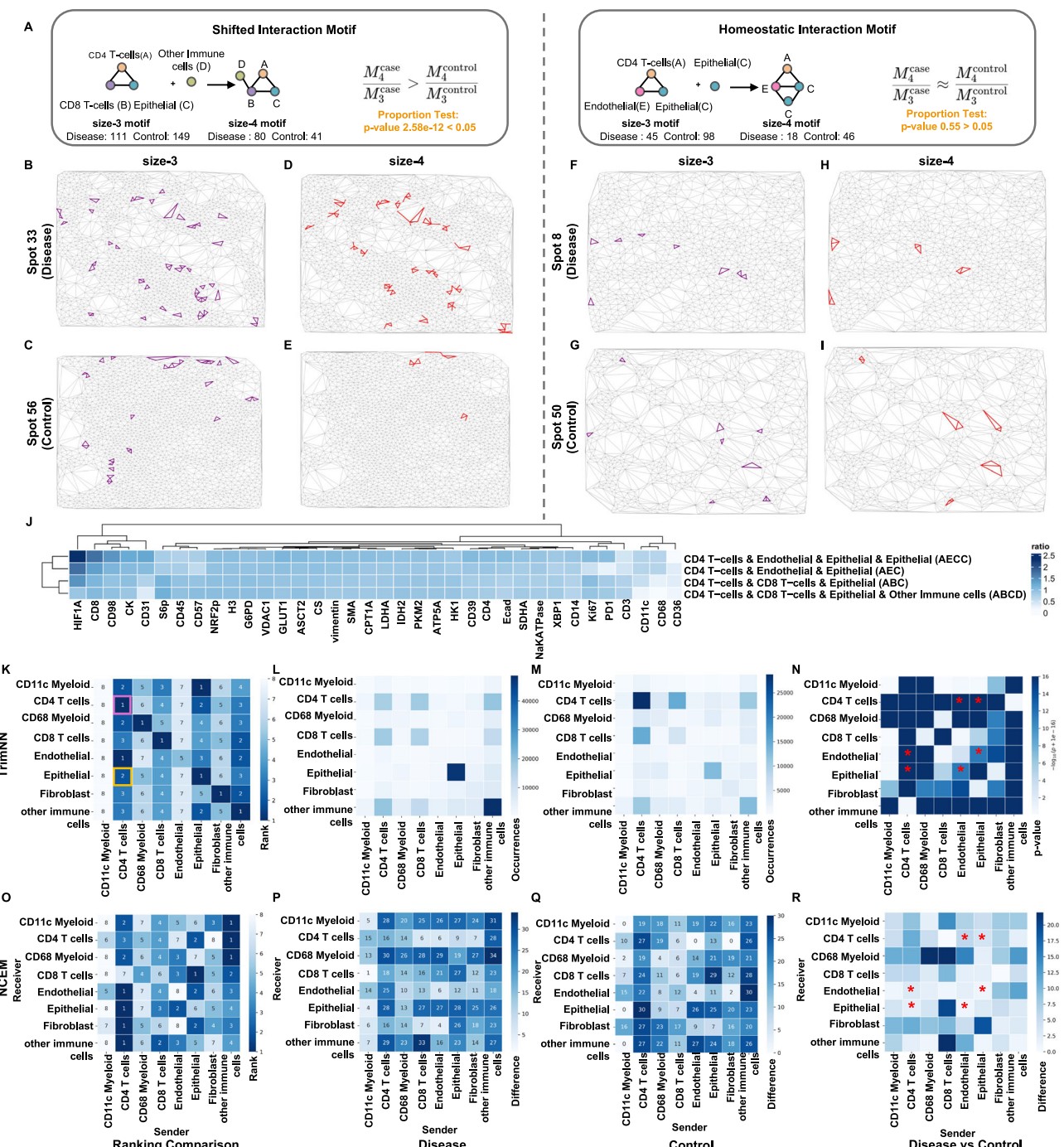

**Fig. 5 | TrimNN analysis in a colorectal carcinoma study using MIBI-TOF.**
**A** Schematic of Shifted Interaction Motif and Homeostatic Interaction Motif as two types of size-4 motifs. Shifted Interaction Motifs: size-3 motif "ABC" (purple) in exemplary **B** spot 33 (disease) and **C** spot 56 (control), the successor size-4 motif "ABCD" (red) in the same **D** spot 33 (disease) and **E** spot 56 (control). Homeostatic Interaction Motifs: exemplary size-3 motif "AEC" (purple) in exemplary **F** spot 8 (disease) and **G** spot 50 (control), the successor size-4 motif "AECC" (red) in the same **H** spot 8 (disease) and **I** spot 50 (control). **J** Heatmap of antibody expression ratio between disease and control samples in Shifted Interaction Motif and Homeostatic Interaction Motif. **K** Ranking of effective size between all cell types in colon tissue samples. Abundance of size-2 CC motifs as occurrences in **L** colorectal

carcinoma and **M** healthy control samples. **N** P-value of size-2 CC motifs between disease and control by the two-sided Benjamini-Hochberg adjusted Fisher's exact test. **O** Heatmap on sender rank from NCEM-type coupling analysis in colon tissue samples. Heatmap of NCEM-type coupling analysis in **P** colorectal carcinoma and **Q** healthy control samples. **R** Difference values from NCEM-type coupling analysis between disease and control samples. Cell type "A" denotes CD4 T-cells, "B" denotes CD8 T-cells, "C" denotes epithelial, "D" denotes other immune cells annotated by the original publication, and "E" denotes endothelial. The star symbol marks the paired cell type composition of the "AEC" motif. CC: cellular community. Source data are provided as a Source Data file.

other immune cells to form a size-4 motif "ABCD" (Fig. 5D, E), where "A" denotes CD4 T-cells, "B" denotes CD8 T-cells, "C" denotes epithelial, and "D" denotes other immune cells (other CD45+) annotated by the original publication. Proportion tests showed that this size-4

"ABCD" motif significantly differed from the size-3 "ABC" motif in abundance between disease and control samples ($p = 2.58e{-}12$). In contrast, Homeostatic Interaction Motif remained consistent in abundance between disease and control groups when expanding its

sizes (e.g., size-3 motif "AEC") (Fig. 5F, G), where "E" denotes endothelial concatenating another epithelial ("C") as a size-4 motif "AECC" (Fig. 5H, I). Proportion tests showed consistency between disease and control ratios among this pair of size-3 and size-4 motifs ($p = 0.55$). The expression level of antibodies also confirmed differences between these two groups of CC motifs (Fig. 5J, Supplementary Fig. 35A and 35B). A similar analysis demonstrated that these two groups of CC motifs also existed in AD studies (Supplementary Notes 2).

To better explain the biological relevance of these patterns, we analyzed the types of proteins showing increased expression in the expanded motifs. Interestingly, the increased disease/control fold-change observed in "ABCD" compared to "ABC" suggests that the involvement of the fourth node (other immune cells) is associated with heightened immune activity or immune cell engagement in the tumor microenvironment. Many of the upregulated proteins in this motif, such as CD39, CD11C, CD45, CD14, and H3, are related to immune infiltration, antigen presentation, or immune cell function. However, the presence of CD39 (involved in immunosuppressive adenosine signaling) and a concurrent rise in metabolic enzymes (e.g., HK1, G6PD, and VDAC1) and proliferation markers (e.g., Ki67) suggests that this immune activity may be skewed or dysfunctional. Rather than a classical immune-activated phenotype, the tumor microenvironment in "ABCD" may represent an immune-infiltrated but metabolically constrained immunoregulatory niche.

In contrast, although the Homeostatic Interaction Motifs "AEC" and "AECC" do not change in abundance across conditions, their protein expression reveals different biological characteristics. Notably, PD1 and GLUT1 are upregulated in the size-4 motif "AECC" suggesting that immune checkpoint pathways and metabolic competition may play important roles even in spatially stable interaction contexts. PD1 upregulation indicates a checkpoint-mediated immune inhibitory signal, while GLUT1 elevation supports increased glycolytic activity, both of which can further restrict immune function through resource competition. We also observed that motif-specific differences in PD1 and CD3 did not change between "ABC" and "ABCD", but both showed increased expression in "AECC" versus "AEC". This implies that the involvement of endothelial cells may contribute to a niche promoting immune checkpoint activation or T cell exhaustion. CD36, a lipid metabolism regulator, showed a moderate decrease in the "ABCD" motif, suggesting lipid metabolic shifts associated with certain spatial configurations.

Together, these comparisons provide evidence that different motif expansions not only reflect structural reorganization but also correspond to distinct biological states. The Shifted Interaction Motif represents a transition toward a metabolically active, immune-suppressive tumor niche, consistent with tumor progression and immune evasion. The Homeostatic Interaction Motif, while stable in structure, still shows signs of immune checkpoint engagement, potentially limiting effective immune surveillance.

Next, we explored cell-type preferences using TrimNN analysis and identified CD4 T-cells that played key spatial roles in differentiating patients and healthy controls. Scrutinizing spatial co-localizations between cells in different cell types, homogeneous CD4 T-cells were most abundant in healthy control samples but ranked fifth in disease samples (purple rectangle in Fig. 5K) by effective sizes, with the $p$-value $< 10e{-}30$, using the Benjamini-Hochberg adjusted Fisher's test (Fig. 5L, M and Supplementary Data 64). This result was also supported by differences in the sender–receiver effect by NCEM[47], which estimates cell interactions of the effects of niche composition (Fig. 5O–Q). In addition, CD4 T-cells and epithelial cells tended to be more likely to be localized together than other combinations (yellow rectangle in Fig. 5K). This combination was also significantly differentiated between disease and control (adjusted $p$-value $< 10e{-}30$), supported by the observation that epithelial cells as the receiver

significantly differ in antibody expression in CD4 T-cells, sender effect (Supplementary Fig. 35C, D) from NCEM. Statistical coupling analysis within all the cell types between disease and control samples (Fig. 5N) showed that many were consistent with NCEM-type coupling analysis (Fig. 5R). All these observations were consistent with the original studies[46].

We also investigated the motifs with sizes larger than 2 where NCEM cannot be inferred directly. The abundance of CC motif "AEC" ("CD4 T-cells & Endothelial & Epithelial") was observed to be significantly different ($p = 4.61e{-}15$) between disease and control (Supplementary Data 65), where their composed pairs of cell types as size-2 motifs were confirmed as significant by TrimNN (Fig. 5N). However, NCEM failed to support this observation, for none of these pairs were shown to be significantly different (Fig. 5R marked as red "*"). Based on these analyses, TrimNN also found that motif "AEC", a Homeostatic Interaction Motif, concatenating another cell type, epithelial, as a size-4 motif, was significant in disease and control. Similar analyses were performed to identify Shifted Interaction Motifs in Supplementary Notes 3. These results were also validated by an independent colorectal carcinoma study[48] using CODEX. The Homeostatic Interaction Motif remained consistent in abundance between the disease and control groups during the growth from size-3 to size-4, which was consistent with our observations (Supplementary Data 67 and 68). Using TrimNN, we recognized that the decline of dominant CD4 T-cells in spatial space may be linked to colorectal carcinoma where CD4 T-cells had more spatial relations with other cell types[49].

## Discussion

Spatial omics has significantly advanced our understanding of the nuanced cell organization within tissues at the cellular level. Being complementary to top-down approaches, such as clustering methods, TrimNN enables the characterization of sizable CC motifs and provides a new bottom-up angle in spatial omics analysis. (1) It overcomes the limitation of clustering approaches. Without arbitrary parameters in clustering, the bottom-up approach identifies topological building blocks as countable CC motifs to represent CNs. (2) It provides intrinsic interpretability of the topological building blocks of CNs. Easily interpretable biologically at the cellular and gene levels, and pathologically intertwined with clinical phenotypes, the results of TrimNN demonstrate the co-localization differences between repeated motifs and disease markers, which may correspond to different biological and pathological hypotheses. (3) It ensures better generalizability across samples. Without a complex machine learning model with uninterpretable embeddings dependent on trained samples, CC motifs as explicit interpretable representations are robust in differentiating CNs and noises.

In many ways, the relationship between clustering-based approaches and CC motifs is similar to that between functional domains (such as Pfam[50]) and sequence motifs (such as PROSITE[51]) in protein research. For example, a typical protein sequence motif is a short, conserved amino acid pattern, often 10 to 20 residues in length, exemplified by motifs like the Zn-finger. These motifs are particularly useful for identifying fine-scale evolutionary differences, such as the gain or loss of protein functions resulting from motif-level changes. Functional domains represent larger, conserved sequences functioning as independent structural and functional units, providing broad functional annotations. In spatial biology, like protein functional domains, clustering provides a top-down view of tissue organization by identifying broad groupings of cell types based on composition and co-localization. In contrast, CC motifs, like protein sequence motifs, offer a bottom-up, fine-grained perspective by capturing conserved topological patterns of cell interactions. While clustering reveals high-level structural features, CC motifs uncover subtle, biologically meaningful differences, such as those between healthy and diseased

tissues, enhancing sensitivity in comparative analyses and enabling deeper mechanistic insights.

Mathematically, TrimNN offers an accurate, unbiased, efficient, and robust approach to quantifying CC motifs as interpretable building blocks of cell organization. The proposed work formulates the pattern quantification problem in counting subgraph occurrences, simplifying the task to a biologically constrained problem that can be solved by a supervised topological representation learning framework using triangles as the inductive bias. TrimNN's effectiveness depends on its ability to (1) simplify the NP-complete subgraph matching problem on the universal graph to a well-defined set of biologically meaningful triangulated graphs; (2) decompose the challenging isomorphic counting regression problem on the entire graph into many straightforward binary present/absent prediction problems on small graphs, which makes it possible to estimate biologically meaningful top overrepresented CC motifs with relative values; (3) use PE-based graph representation methods to empower the expressive power of the model in spatial omics; and (4) use the supervised learning approach to move the computational tasks to the training process to accelerate the inference to facilitate users' practice. This work paves the way for disclosing the biological mechanisms underlying multicellular differentiation, development, and disease progression.

Biologically, TrimNN opens new opportunities for discovering complex mechanisms in complex diseases using spatial omics. The idea of CC motifs can be treated as an explicit representation that simplifies the cell organization and preserves the topological relations in the context of triangles derived from Delaunay triangulation, a fast and reliable mathematical process on spatial spaces as a CC. Using an interpretable LR model in the CRC study, the results of TrimNN robustly differentiate phenotypical and pathological characteristics of spatial omics samples. With TrimNN, we identified diverse CC motifs in spatial localization and cellular, gene, and pathway features in AD and colorectal carcinoma studies, corresponding to diverse biological and pathological mechanisms in complex diseases, which were often overlooked by conventional analysis.

Furthermore, finding CC motifs is also related to mining FTUs at the atlas level. For example, identifying tumor lysis syndrome[4] in CNs in cancer research helps to illustrate how the immune microenvironment plays a role in cancer progression. Using topology-based cell type combinations in the spatial space, the abundance of CC motifs describes the dispersed and coherent cell organizations as shown in a demo system in Supplementary Fig. 36. In complex systems such as the tumor microenvironment, these abundances can be used to describe the cell type boundary and gradient. Especially, the abundance of specific cell types of tumor-immune mixing is also known to distinguish different organization scenarios, such as a cold tumor, mixed tumor, and compartmentalized tumor, which are directly related to survival in triple-negative breast cancer[52].

Finally, we note that both the theory and practice of determining the optimal spatial scale for describing observed data in spatial biology remain in their early stages. We comprehensively investigated the relationship between motif size and its impact on performance in Supplementary Note 4, which includes experiments exploring the robustness simulation to size-5 (Fig. 2A–C), size-12 in the context of AD research (Supplementary Figs. 37–39 and Supplementary Data 69–71), and size-7 motifs in a demo system (Supplementary Fig. 40 and Supplementary Data 72). From the limited experiments and observations, there may not be one optimal size of CC motifs that uniformly describes CCs within the complex biological context revealed from the spatial omics. The selection of an optimal motif size requires extensive statistical studies, which will be our future work. For example, if a reliable approach can be established to assess the p-value of each motif between case and control, the motifs with the smallest p-values should be used as motif markers.

In practice, there are trade-offs between large and small sizes of CC motifs in presenting the characteristics of CNs in spatial biology. (1) Complex large-size CC motifs may demonstrate better prediction power, but they are hard to identify. In the case study of CRC, we confirmed that size-3 CC motifs perform better in classification than size-1 CC motifs. However, the number of motifs increases exponentially with motif size, which makes identifying the most overrepresented motifs in large sizes computationally expensive. (2) CC motifs in large sizes have similar biological interpretability to smaller sizes but lack an established statistical framework in scientific rigor. There are no limits on size in order to use biological interpretation approaches on these motifs. However, to the best of our knowledge, no single statistical test is comparable to Fisher's exact test (or the chi-squared test) or the CMH test that universally applies to four-way or higher-dimensional contingency tables. (3) More caution needs to be taken when applying complex large-size CC motifs in machine learning because of overfitting. When applying these motifs to current spatial omics datasets with a limited number of samples, users need to be aware of selecting the appropriate number of top CC motifs as features in order to perform machine learning tasks to prevent overfitting.

This work still has some limitations. First, TrimNN enumerates the possible graph topologies and greedy search strategy to identify the large-size motifs, which has room for improvement with deep reinforcement learning approaches. Second, TrimNN still estimates the relative rankings in abundance and cannot be guaranteed to be exact. Last, current settings in CCs may oversimplify the problem without any features on the edges and nodes. We may add essential features (e.g., morphology features) for specific applications. Future work will include analyzing large-scale spatial omics data to connect the idea of CC motifs and FTU, developing systematic evaluation metrics in multiple sizes of CC motifs, and assessing the results in different categories of diseases from multiple independent data sources.

## Methods
### Problem setting

Formally, we define the triangulated graph $G$ as the CC inferred from spatial omics using Delaunay triangulation[13], where $G = \{V, E\}$ with $|V| = n$ nodes and $|E|$ edges. The size-$k$ CC motif is a subgraph with $k$ nodes as $m_k$, where $m_k$ is an induced subgraph of $G$. Here, $m_k = (V', E')$ is defined as an induced subgraph if and only if when $V' \subseteq V$ and $E' = \{(u, v) \in E | u, v \in V'\}$. $M_k$ is the set of all $m_k$ of size-$k$, and $M_k \subseteq G$. The biological problem of identifying the overrepresented CC motifs can be modeled mathematically in finding the most overrepresented subgraph $m_k^*$ in $G$, where $m_k^* \subseteq G$ and $m_k^* \in M_k$. This challenge consists of a subgraph matching problem and a pattern growth problem built on it.

TrimNN aims to address the subgraph matching problem in the context of spatial omics, which seeks to define a function $F(G, m_k) \in \mathbb{N}$, estimating the relative occurrence of the given $m_k$ in $G$. In our setting, this problem can be semi divided and conquered by summarizing many sub-TrimNN problems. The goal of sub-TrimNN is to build a reliable binary prediction model $f(g, m_k) \in [0, 1]$, where 0 represents that $m_k$ is absent in graph $g$ and 1 represents presence, $g \subseteq G$. With sub-TrimNN on enclosed graphs centered by each node, TrimNN is the summarization of results from all sub-TrimNN in the graph, as Eq. (1):

$$F(G, m_k) = \sum_{v \in G} f(g(v, hop), m_k) \tag{1}$$

where $g(v, hop)$ is the enclosed graph as the neighborhoods of node $v \in V$ with $hop \in [1, 2, 3, \ldots]$, and $g(v, hop) \subseteq G$. Generally, the value of $hop$ is related to the length of the longest path of $m_k$. Here, we use $hop = 2$ in all the analyses to make enclosed graphs $g$ is in a similar size of $m_k$.

We use a fast and reliable $F(G, m_k)$ from TrimNN to address the problem of pattern growth. Using searching processes, the goal is to get a top overrepresented set $m_k^*$ has the maximum relative abundance in Eq. (2):

$$F(G, m_k^*) = \max_{m_k \in M_k} \left( F(G, m_k) \right) \qquad (2)$$

If both case and control samples are available, $F\prime(G_{case}, G_{control}, m_k)$ is defined to find CC motif $m_k$ that differentiates $G_{case}$ and $G_{control}$ in Eq. (3), where $G_{case}$ represents CC from case samples and $G_{control}$ represents CC from control samples. $F\prime$ can be any function to describe the differences, including Fisher's exact test or effective size. $m_k^*$ is the top overrepresented set mostly differentiated conditions in Eq. (4).

$$F\prime(G_{case}, G_{control}, m_k) = F'\left( F(G_{case}, m_k), F(G_{control}, m_k) \right) \qquad (3)$$

$$F\prime(G_{case}, G_{control}, m_k^*) = \max_{m_k \in M_k} \left( F'(G_{case}, G_{control}, m_k) \right) \qquad (4)$$

## TrimNN in subgraph matching

We decomposed the regression problem of TrimNN in the CC into many binary classification problems in enclosed graphs centered by each node of the triangulated graph. This classification problem on each enclosed graph is solved by sub-TrimNN. The input of sub-TrimNN is a pair of subgraphs of query $m_k$ and the target triangulated graph $g$. To better represent the topological information of subgraphs and graphs, we use an empowered GNN based on GIN and a shortest distance positional encoding $PE$, which can be denoted as Eq. (5):

$$h_v^{(l)} = \text{MLP}^{(l)}((1 + \varepsilon^{(l)})^* h_v^{(l-1)} + \sum_{u \in N(v)} h_u^{(l-1)} + a^*PE) \qquad (5)$$

where $h_v^{(l)}$ is the learned embedding of node $v$ at the $l$th layer. MLPs are multi-layer perceptrons, $\varepsilon$ is a fixed scalar and $N(v)$ is a set of nodes adjacent to $v$, and $a$ is the scaling factor for controlling the strength of $PE$. Here, $PE$ as the shortest distance positional encoding[24] is adapted to encode the relative positions of nodes in a graph based on the shortest path distances between them, described as Eq. (6):

$$PE = \left[ d(v_0, v_0), d(v_0, v_1), d(v_0, v_2), \dots, d(v_0, v_n) \right] \qquad (6)$$

where $d(u, v)$ denotes the shortest distance between two nodes $u$ and $v$. For all nodes $v$ in the graph $g$, we first selected one endpoint $v_0$ of the shortest path in the whole graph $g$ as the starting point. Then the shortest distance from this point was calculated for all other vertices. After obtaining $PE$ for the paired query $m_k$ and target $g$, we extended the positional encoding of each node to match the dimensionality of the learned graph embeddings from GIN and then added them together. Then we passed learned node representations through a graph max pooling layer[53] to get the graph representations. After the sigmoid function activates linear layers, sub-TrimNN outputs the binary predictions: 1 as presence and 0 as absence. The whole training process aims to minimize the cross-entropy loss function of known presence/absence relations as Eq. (7):

$$L = -\frac{1}{N} \sum_{i=1}^{N} \left[ y_i \log(\hat{y}_i) + (1 - y_i) \log(1 - \hat{y}_i) \right] \qquad (7)$$

where $N$ is the number of samples, $y_i$ is the true label for the $i$-th sample (0 or 1), and $\hat{y}_i$ is the predicted probability for the $i$-th sample. After trained sub-TrimNN $f(g, m_k)$, TrimNN estimates the abundances of $F(G, m_k)$ by summarizing sub-TrimNN predictions on each node's enclosed graph as Eq. (1).

Theoretically, both the time and space complexity of sub-TrimNN inference are $O(|V| + k)$, which is linear to the node sizes of the input subgraph and the triangulated graph. The time complexity of the entire TrimNN inference is the graph node size multiples the sub-TrimNN on all the enclosed graphs, which is $O(|V|^*(hop^*k + k))$, and the time complexity of building an enclosed graph is $O(|V|^{hop})$. The space complexity of the entire TrimNN inference is $O(|V|^{hop^*}(hop^*k + k))$, where $hop$ is defined in generating the enclosed graph. On the other hand, the space complexity of VF2 is of order $O(V)$, and the time complexity is $O(V!^*V)$.

## Greedy strategy in pattern growth

In the pattern growth process, we use the function $F(G, m_k)$ from the subgraph matching in Eq. (2) to find $m_k^*$ for a specified size $K$ with a serial of greedy search. Starting from small sizes of CC motifs of size-$k$, where $k = 1, 2$, or $3$, we enumerate all possible subgraphs $m_k$ and then obtain their corresponding predicted occurrence values using trained $F(G, m_k)$. For studies with case and control conditions, we calculate the $p$-value of Fisher's exact test to identify the most significant CC motifs between different conditions. If no case–control information is available, we select the subgraph with the maximum relative abundance as $m_k^*$. After obtaining $m_k^*$ at size-$k$, we use it as a seed to enumerate all possible size-$k + 1$ subgraphs $M_{k+1}$ based on $m_k^*$.

$$M_{k+1} = \bigcup_{v \in N(m_k^*), e \in E(m_k^*)} \{m_k^*, v, e\} \qquad (8)$$

where $E(m_k^*)$ is a set of edges linked to the graph $m_k^*$. Each node type as a new node is selected with $m_k^*$ to get a new size-$k + 1$ graph. Similar to Eq. (1), $m_{k+1}^*$ is defined as Eq. (9) in $M_{k+1}$.

$$F(G, m_{k+1}^*) = \max_{m_{k+1} \in M_{k+1}} \left( F(G, m_{k+1}) \right) \qquad (9)$$

If both case and control samples are available, $m_{k+1}^*$ is defined as Eq. (10) in $M_{k+1}$.

$$F'(G_{case}, G_{control}, m_{k+1}^*) = \max_{m_{k+1} \in M_{k+1}} \left( F'(G_{case}, G_{control}, m_{k+1}) \right) \qquad (10)$$

By iterating the process of Eq. (8) with incremental $k$, we can find $m_k^*$ at specific size $K$ in Eq. (9) or Eq. (10) if case and control samples are available, where $K \in \mathbb{N}^+$.

## Constructing the training dataset

In spatial omics samples, the spatial relations between the cells can be modeled as a CC in a cell graph using the Delaunay triangulation[13] on their spatial coordinates. In the generated cell graph composed of triangles after triangulation, each node denotes a cell and is labeled with a cell type, and each edge represents a hypothetical spatial relation between two cells.

We build a comprehensive synthetic training set with ground truth presence/absence relations between pairs of the query CC motif (subgraph) and the target triangulated graph. The classical tool VF2[21] generated the ground truth occurrences by enumerating all the possibilities and guaranteeing the exact results with a substantial computational cost. To preserve biologically meaningful diversity, we simulated 7 distinct CC motifs from size-3 to size-9 in various topologies (Supplementary Fig. 41). Given the context of routine spatial omics in ROIs for each CC motif, we constructed the corresponding triangulated graphs with varying node types of 8, 16, and 32. The node types were randomly assigned to one of the cell types in the CC under a uniform distribution. To simulate different sizes of CC motifs corresponding to varying sizes of target graphs, we generated triangulated graphs of size-16 and size-32 for size-3 to size-6 CC motifs, and triangulated graphs of size-32 and size-64 for size-7 to size-9 CC motifs.

Each pair of the query and the target has the same number of node types. To preserve the diversity, we generated 50 extended subgraphs with permuted node types for each CC motif. In total, we generated corresponding 1000 triangulated graphs permuting node types for each extended subgraph. To ensure a balanced ratio of positive and negative relations in presence and absence, we controlled the proportion of positive to negative samples at 1:1 in data generation. We split and set the generated data into training, validation, and testing sets in a ratio of 8:1:1. Noteworthily, to fairly test model performance and objectively evaluate the model's actual generalization ability, we constructed an independent test set. For each type of CC motif, we selected 50 entirely new permutations and generated 100 triangulated graphs for each permutation.

### Evaluation performance on synthetic data
We selected CC motif sizes ranging from size-3 to size-9 in the generated synthetic data to demonstrate the power of TrimNN with ground truth. As a binary prediction task of subgraph matching in the synthetic dataset, the performance was evaluated by precision, recall, F1 score, and MCC on the generated test set of varying sizes and node types. For NSIC regresses continuous occurrence, we treat NSIC's prediction of 0 as the query absent in the target graph, and any value larger than 0 as present in the target graph. We omitted VF2 in the performance comparison, for it has already been used to generate the ground truth at a substantial computational cost.

To evaluate the quantitative performance in the synthetic dataset, we used metrics related to recommendation tasks, such as RMSE and mean absolute error (MAE), to quantify the accuracy of the predicted occurrences. We selected three distinct CC motifs, one in size-3, one in size-4, and one in size-5 (Supplementary Data 2 and 3). Then we enumerated all the possibilities in both 8 cell types and 16 cell types with VF2 using significant computational resources. Because we value the biologically meaningful top overrepresented CC motifs in practice, we highlighted whether these methods can successfully identify the top 5 and top 10 overrepresented candidates as CC motifs.

In scalability analysis, the query subgraph contains 9 nodes, and both the subgraph and the triangulated graph have 32 node types. All the experiments were performed on a workstation equipped with an Intel Xeon Gold 6338 CPU with one NVIDIA A100 GPU and 80 G RAM.

### Data preprocessing
**Constructing triangulated graphs.** We applied Delaunay triangulation to construct the cellular community based on the localization of cells with spatial coordinates. We added an optional step for noise reduction with unnecessary edges to prevent potential artifact edges from being included in certain boundary regions. We computed the lengths of all edges and used the 99th percentile as a threshold to remove outlier edges that were too long.

**Processing spatial proteomics in colorectal cancer study.** The CRC study[13] included in the analysis uses the spatial proteomics approach CODEX at the single-cell resolution, which contains 140 tissue regions from 35 advanced-stage CRC patients with 56 protein markers and 29 distinct cell types. This study consists of 17 patients labeled as "Crohn's-like reaction" (CLR) and the remaining 18 patients as "diffuse inflammatory infiltration" (DII). Each patient has CODEX data with two spots, and each spot has two regions. Coordinates and cell type annotations of the cells were from the original publication. For all 140 samples, the summarized occurrence numbers of size-1 to size-4 CC motifs were used in the classification tasks to predict the corresponding patients in either CLR or DII. Here, we chose a unified number 29 as the number of features in size-2 to size-4 CC motif analysis, in alignment with the size-1 29 features as the total number of cell types. To get the top size-4 motifs, we performed TrimNN based on the top 29 size-3 CC motifs as triangles.

### Processing spatial transcriptomics in an Alzheimer's disease study.
In the AD datasets using STARmap PLUS spatial transcriptomics in this study[34], 8 AD samples of mouse brain tissues—two replicates of an 8-month-old AD and control, together with a 13-month-old AD and control—were utilized for analysis. The original study provided each cell's coordinates and its cell type annotation. The CC was built using Delaunay triangulation, size 1 to size-3 CC motifs were identified by enumeration, and the size-4 CC motif was inferred through TrimNN. In the downstream analysis on the cellular level, we took only the disease samples with amyloid-β regions and combined replicates 1 and 2. We categorized the samples' regions containing specific CC motifs and their extended 3-hop as motif regions, and all the remaining regions were complementary non-motif regions. Then we performed the cell–cell communication analysis using CellChat and DEG analysis using the Wilcoxon rank-sum test. The spatial localization of amyloid-β and cells' spatial coordinates were used to compute the shortest distance from amyloid-β to motifs.

### Processing spatial proteomics in colorectal carcinoma study.
This study[46] uses MIBI-TOF with 36 antibodies on colorectal carcinoma and comprises 58 ROIs within the spatial information and expression at the proteomics level. 40 ROIs were from two disease patients, and 18 ROIs were from two control patients. Both NCEM and TrimNN were used to analyze these 58 ROIs.

### Noise simulation
To further validate the robustness of CC motifs, we applied three types of simulations based on STARmap PLUS data in the AD study[34]. These simulations manually added noise to mimic limitations arising from sequencing technologies and data processing in practical analysis, including (1) cell missing, (2) cell coordinate shifting, and (3) cell type misclassification.

(1) Cell missing targets to mimic the limited sequencing capacity to identify cells in the spatial omics samples. Proportions of cells in the input at rates of 0.01, 0.05, 0.1, 0.2, and 0.5 were randomly removed.

(2) Cell coordinate shifting aims to mimic errors in sequencing or shifting in sample preparation to identify the spatial localization of the cells. For proportions of cells at rates of 0.01, 0.05, 0.1, 0.2, and 0.5, their coordinates were shifted proportion of 0.01, 0.05, 0.1, 0.2, and 0.5 with their average distances to the nearest neighboring cells in random directions.

(3) Cell type misclassification aims to mimic errors in annotating cell types from spatial omics samples, possibly due to insufficient cell type annotations. In the simulation, original cell types were randomly shuffled in cells at proportions of 0.01, 0.05, 0.1, 0.2, and 0.5.

To test the robustness of CC motifs in abundance rankings, all the simulations were randomly generated 100 times. Spearman correlation was adopted to compare the abundance ranks of all CC motifs between the original and the noisy datasets.

To test the generalizability of machine learning models, random cropping was proposed in addition to noises from cell missing, cell coordinate shifting, and cell type misclassification. Random cropping aims to simulate smaller ROIs due to limited sequencing capacity or incomplete sample preparation, in that only a portion of the original tissue is captured. In this simulation, new patches as part of the original samples were randomly cropped with 0.1, 0.2, 0.3, 0.4, 0.5, 0.6, 0.7, 0.8, and 0.9 proportions of the original width and height. Random cropping was performed 10 times in each parameter setting. As part of the original samples, these newly generated small patches had identical phenotypes and survival of the original data. In generalization analysis, the machine learning model was trained on all the original CRC data and tested on the distorted samples.

### Statistics summary
**False discovery rate (FDR).** To assess the significance of CC motifs across samples, Fisher's exact test and Chi-squared test were adapted

in the study, and they were adjusted using Bonferroni correction and the Benjamini-Hochberg method.

**Effect size.** In large sample sizes, even small differences can become statistically significant (i.e., the *p*-value can be very small), which might not be practically significant. Therefore, we used the Cramér's V effect size with a moderate effect size threshold of 0.21 in addition to the *p*-values to determine the significance of the Chi-squared tests[54]. For the hypothesis test that the occurrence of cell type "A" is independent of the occurrence of cell type "B", the effect size of the chi-squared test is in Eq. (11):

$$ES_{AB} = \sqrt{\frac{\chi^2}{n \cdot (k-1)}} \tag{11}$$

where $\chi^2$ is the chi-squared statistic, $n$ is the total number of edges involved in each test, and $k$ is 2 for the two-by-two contingency table. For the triangle of three types of cells in size-3 CC motifs, denoted as "A", "B", and "C", we fixed one type of cell in the triangle and calculated the effect size for the test between the other two types of cells as in Eq. (12):

$$ES_{AB|C} = \sqrt{\frac{\chi^2}{n_C \cdot (k-1)}} \tag{12}$$

where $n_C$ denotes the total number of triangles that include at least one cell of "C", and $k$ is 2. The overall effect size was determined as the minimum value of $ES_{AB|C}$, $ES_{AC|B}$, and $ES_{BC|A}$, i.e., $ES_{ABC} = \min(ES_{AB|C}, ES_{AC|B}, ES_{BC|A})$, to ensure that the effect size of each chi-squared test is above the threshold.

**Proportion test.** We performed a two-sample, two-sided proportion test to test the significance between size-3 CC motifs and size-4 CC motifs. The null hypothesis ($H_0$) was that the proportions were equal between the two groups (i.e., $H_0$: the proportion of size-4 motifs within size-3 motifs is the same in both the disease and control groups).

### Parameter settings
**Phenotypic classification in supervised learning.** The top abundant CC motifs in size-1 to size-4 were selected as the features to represent CNs. Specifically, the dense rank of the occurrence was scaled to [0, 1] by the function min_max. Classical machine learning models Logistic Regression, Random Forest, and Support Vector Machine were adopted to perform classification tasks using 10 times tenfold cross-validation following the same protocol as CytoCommunity. Comprehensive criteria such as F1 score, precision, recall, MCC, area under the precision curve, and area under the Receiver Operating Characteristics Curve (ROC-AUC) were used to measure the binary classification performances.

**Survival curve.** In survival analysis, we also summarized the overall occurrence of CC motifs through 140 regions for each size. The definition of whether a CC motif is enriched in a patient is by identifying whether it is among the top 29 motifs of its own size. Kaplan–Meier curves showing survival as a function of time for patients with and without CC motifs enriched in cell organizations using R packages "*survival*" and "*survminer*". The hazard ratio (HR) and *p*-value (P) were calculated using Cox regression analysis.

**Cell–cell communication analysis.** In the study of spatial transcriptomics datasets, CellChat[35] performed ligand–receptor analysis using the "*TruncatedMean*" method suggested by the official tutorial. In the cross-sample analysis, we retrieved the pathways related to targeted cell types of the single cell–cell interaction (targeted cell types as source or as target interacting with all other cell types) if the

difference of cross-sample values is over 1. In the study of spatial proteomics datasets, we performed NCEM[47] type coupling and genewise analysis to supplement our size-2 triangulation analysis. All the parameters followed official tutorials.

**Gene-level analysis.** We used the Wilcoxon rank-sum test in Scikit-learn[55] package to infer DEGs between motif and non-motif regions. Genes with *p*-values less than 0.05 were inferred as DEGs. GO and pathway enrichment analyses were adopted by clusterProfiler[56]. Then we applied the R package "*wilcoxauc*" to perform the Wilcoxon rank-sum test to identify marker genes in different motifs.

**Spatial co-occurrence analysis.** We used Squidpy[41] function "*co_occurrence*" to calculate spatial co-occurrence probability between CC motifs and amyloid-β. The coordinates of CC motifs are the averaged coordinates of each node in the motif.

**Functional evaluation for multisize motifs.** We extended the motif size up to 10 triangles (typically size-12) (shown in Supplementary Fig. 37) and benchmarked the exact occurrences of each motif using the enumerative searching approach VF2 with guaranteed accuracy. For each motif in multiple sizes, we identified its unique DEGs by comparing cells within the motif regions, as identified by VF2, to those outside the regions. GO and pathway enrichment analyses were then performed on these motif-related DEGs. The quality of CC motifs was evaluated and compared by *q*-values of enriched GO terms and pathways. To establish a baseline for AD relevance, we selected the top 300 GO terms found in the intersection of significant terms from four size-3 motifs: "CCC", "CCM", "CMM", and "MMM" in 13-month-old AD disease samples. For each motif, the Cauchy meta-analysis[57] was applied to combine the *p*-values of these selected GO terms, with the resulting -log(*p*-value) as a quantitative measurement of relevance to AD.

**CytoCommunity and Space-GM.** We used CytoCommunity and Space-GM to perform the supervised learning as the benchmark using the default parameter setting. Using the same protocol in CytoCommunity, we used the same 10 times tenfold cross-validation and the same evaluation metrics for all the methods. It is notable that CytoCommunity performs training within in default 20 epochs. The output from the last epoch was shown as the final results.

**DeepTalk.** Following the official tutorial of DeepTalk, we first integrated the STARmap PLUS AD dataset and scRNA-seq data GSE176032 downloaded from Gene Expression Omnibus (GEO). We identified the cell–cell communication score and focused on the ligand–receptor pairs: Ccl3-Ccr5, Csf1-Csf1r, Gdf15-Tgfbr2, Nrg3-Erbb4, Pdgfa-Pdgfra, Pdgfa-Pdgfrb, Pyy-Npy1r, Vegfa-Flt1, and Vip-Vipr2.

### Reporting summary
Further information on research design is available in the Nature Portfolio Reporting Summary linked to this article.

## Data availability
All relevant data supporting the key findings of this study are available within the article and its supplementary information files. The human CRC CODEX dataset used in the colorectal cancer case study is available at https://data.mendeley.com/datasets/mpjzbtfgfr/1. The STARmap PLUS sequencing data in the Alzheimer's disease case study are available at https://zenodo.org/records/7332091. The MIBI-TOF imaging data of colorectal carcinoma and healthy colon in the spatial proteomics case study are available at https://doi.org/10.5281/zenodo.3951613. The generated simulation data and analysis code has been deposited in Figshare database at https://doi.org/10.6084/m9.figshare.27958218. Source data are provided with this paper.

## Code availability

The source code of TrimNN is freely available at https://github.com/yuyang-0825/TrimNN under MIT license. The specific version of the code associated with this publication is archived in Zenodo and is accessible via https://doi.org/10.5281/zenodo.15884334[58].

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

## Acknowledgements

We thank Jiacheng Xie and Abdul Muqeeth for assisting with some of the analyses. This work is supported by National Institutes of Health grants R01DK138504 (to J.W. and Q.M.), R35GM126985 (to D.X.), R01GM152585, P01CA278732, U54AG075931, and P01AI177687 (to Q.M.), U19AG074879, R01AG019771, P30AG072976, U01AG072177, U01AG068057 (to M.Y.), NS121718 (to J-Q.K), R25HG012325 (to K.M.), R21DK140693 (to A.M.), the AnalytiXIN initiative (to J.W.), and the Alzheimer's Association grants AARF-22-722571 (to M.Y.), as well as the Pelotonia Institute of Immuno-Oncology (PIIO) (to Q.M.).

## Author contributions

Conceptualization: J.W., Q.M., and D.X.; methodology: J.W. and D.X.; software coding: Y.Y. and S.W.; data collection and investigation: S.W., Y.Y., and K.M.; data analysis: S.W., Y.Y., J.L., M.Y., K.M., and A.M.; software testing and tutorial: Y.Y. and S.W.; manuscript writing, review, and editing: J.W., Y.Y., S.W., A.M., Q.M., J-Q.K., and D.X.

## Competing interests

The authors declare no competing interests.
