## [Transparent Peer Review file · Nature Communications]

TrimNN: Characterizing cellular community motifs for studying multicellular topological organization in complex tissues

Corresponding Author: Dr Juexin Wang

Version 0:

Reviewer comments:

Reviewer #1

(Remarks to the Author)

This manuscript presents TrimNN, a graph-based deep learning framework that identifies cellular community (CC) motifs from spatial omics data, offering an interpretable, bottom-up approach to analysing multicellular topological organisation. The method demonstrates computational efficiency and biological relevance in Alzheimer's disease and colorectal carcinoma datasets, and has a broad applicability to a variety of spatial datasets. However, while the study appears methodologically sound and potentially impactful, several concerns should be addressed before publication, particularly regarding biological validation, methodological clarity and benchmarking.

1. The idea of describing cellular communities in the form of graph motifs with variable size is powerful and well suited to spatial data. The authors state that their method's computational efficiency allow them to scale up to considerably large motifs. However, in all their analyses they only present insights from size 1-4 motifs, i.e. communities comprising only up to 4 cells. As such, these insights appear rather limited, even if some of them have biological relevance. What about larger motifs? Are they not relevant/informative? What motif size is most biologically meaningful? It would be very interesting if the authors could devise a way to scan through motifs of various sizes and determine the most suitable configuration of CC motifs (of varying sizes) that best describes the observed data in the slides.

2. While I appreciate the question above is a rather difficult task, I would at least be curious to know how the performance metrics of TrimNN change in figures 2 a-c when adding noise if motifs of size >3 are investigated. It appears there may be a limit to the ability to robustly capture these patterns, with the performance decreasing with the increase in the size of the motif and the amount of noise. I find this rather counterintuitive though – could the authors explain why the correlation is worse when introducing noise in larger motif structures?

3. It is not clear how the synthetic training datasets were generated. Specifically:

- How were node type permutations constructed? Were there any constraints to ensure biological plausibility?
- Given that VF2 was used for ground truth, could this introduce biases, particularly in how TrimNN learns to identify motifs?
- Were real-world spatial omics datasets incorporated into the training process, or was it based entirely on synthetic data? If synthetic, how well does it generalise to experimental datasets?

4. Some biological interpretations remain unclear:

- In the Alzheimer's disease dataset, how do the identified motifs compare with previously known functional tissue units (FTUs) or clustering-based spatial patterns? Are these motifs confirming existing biological knowledge, or do they provide new insights?
- The colorectal cancer survival analysis (Figure 3) is promising, but it requires validation in an independent dataset to confirm its robustness. Have the authors explored external datasets or cross-validation strategies?
- In the colorectal carcinoma study presented in Figure 5, I find the changes in CC patterns less biologically interpretable. What are the functional implications of the shifted vs homeostatic interaction motifs? Could these motifs be linked to immune evasion, tumour progression, or response to therapy? If so, additional discussion would help contextualise their significance.

5. The manuscript discusses the limitations of clustering-based approaches but lacks a quantitative performance

comparison with other spatial omics tools. To strengthen the argument for TrimNN's advantages, the authors should provide a direct performance comparison tools such as CytoCommunity, SPACE-GM, and BANKSY on the same datasets, and benchmark TrimNN's biological motif discovery against standard clustering approaches to highlight what novel insights are uniquely captured.

6. While the discussion highlights TrimNN's advantages, it does not fully address potential methodological biases:

- Delaunay triangulation is used to define CCs, but could this introduce artefacts in tissues with irregular or non-uniform spatial distributions?
- The semi-greedy pattern growth strategy is an interesting choice, but alternatively, could reinforcement learning approaches further refine the search process?

7. The authors use DESeq2 for DEGs analysis between motif and non-motif regions. Since DESeq2 is specifically designed for bulk (negative binomial GLM to handle overdispersion) and requires raw counts (which is quite high in bulk data), I'm not sure if this is the best method to be applied in this instance.

Minor point: The manuscript could benefit from proofreading and support by a native English speaker to make the text easier to read and follow and correct some grammatical errors.

(Remarks on code availability)

I have not tested the code myself, but have reviewed the github repository and it is very well organised and documented. It appears very straightforward to run, with a very clear input that is easily generated from spatial omics data.

Reviewer #2

(Remarks to the Author)

Yu & Wang et al. developed a cellular community (CC) motif discovery tool named TrimNN, designed to identify and quantify the abundance of k-mer CC motifs in spatial omics (ST) data. By leveraging a graph-based deep learning approach, TrimNN enables the detection of conserved multicellular spatial patterns across different biological contexts. The authors validate TrimNN through three case studies:

- (1) Colorectal cancer (CRC), where TrimNN successfully distinguishes patient subtypes (Crohn's-like lymphoid reaction (CLR) vs. diffuse inflammatory infiltration (DII)) based on CC motif abundance, revealing potential prognostic biomarkers.
- (2) Alzheimer's disease (AD), where CC motifs associated with microglia and excitatory neurons are linked to β -amyloid deposition and disease progression.
- (3) Colorectal carcinoma tumor microenvironment (TME), where TrimNN identifies distinct immune-epithelial interaction motifs, highlighting changes in immune cell spatial organization between healthy and tumor tissues.

The study presents an innovative computational framework with potential applications in spatial biology, biomarker discovery, and disease characterization. However, several critical aspects require further clarification and validation, as outlined below:

1. One major concern is that the entire method relies on the topological connections of locally neighboring cells, potentially overlooking long-range cellular communication. This raises the question of whether the identified k-mer CC motifs primarily reflect basic spatial arrangements of cellular composition rather than capturing higher-order functional organization. For instance, the authors construct 4-mer motifs by growing patterns from 3-mer motifs, assuming that the local expansion sufficiently captures biologically meaningful motifs. However, long-range cellular interactions (or long-distance communication) could play a crucial role in tissue organization and function (see: <https://www.nature.com/articles/s41467-024-51329-2>), which this method might fail to capture. To address this limitation, the authors should provide strong biological evidence to support the assumption that long-range interactions are not significantly influential in defining meaningful CC motifs. Alternatively, they should discuss potential methodological extensions that incorporate long-range interactions, such as weighted cell-cell interaction graphs based on cell communication networks (e.g., CellChat), or incorporating graph attention mechanisms to learn distant dependencies (e.g., DeepTalk).

2. In Figure 3A, the authors compared their approach with CytoCommunity (dim = 512) on a dataset of only 35 advanced colorectal cancer (CRC) patients (17 with CLR vs. 18 with DII). A major concern is the discrepancy in feature dimensionality: CytoCommunity used 512-dimensional features, which far exceeds the number of samples (35), raising concerns about overfitting and model instability. In contrast, TrimNN used at most 100 CC motifs (or even as few as 29 motifs), making the comparison potentially unfair. The authors should clarify why a 512-dimensional feature space was used in CytoCommunity, given the small sample size. Furthermore, they should discuss how TrimNN's lower-dimensional motif representation avoids overfitting, particularly in small datasets.

3. In Figure 1B, the ROC curves give the initial impression that larger k-mer CC motifs lead to improved ROC values, suggesting that higher-order motifs provide stronger predictive power. However, in Figure 3a, this pattern does not seem to hold consistently. This raises the question of whether increasing motif complexity (higher k values) always enhances model performance, or if there are diminishing returns or trade-offs associated with using high-order CC motifs. The authors should provide a more detailed discussion on the advantages and disadvantages of high-order complex CC motifs, particularly in terms of predictive power, interpretability, and potential overfitting risks in smaller datasets. A deeper analysis of when and why higher-order motifs improve or fail to improve performance would strengthen the biological and computational justification for TrimNN's motif-based approach.

4. In Figure 4C and 4D, the spatial distribution of the 'MMM' CC motif shows a clear difference between the 13-month-old AD group and the control group, with microglia clustering more tightly in the AD group. However, in Supplementary Figure 8, there is no noticeable difference in the abundance of the MMM CC motif between the 8-month-old AD and control groups. This does not clearly verify whether the 'MMM' motif emerges in the early stages of AD. Moreover, the lack of a larger sample size for statistical analysis makes it difficult to ensure the stability of the 'MMM' motif across individual mice.

5. In Figure 5, the authors identified two types of CC motifs: the Shifted Interaction Motif and the Homeostatic Interaction Motif, which were determined based on the ratio change between size-3 and size-4 motifs. Although the statistical results show significant trends, the study is based on only 2 CRC patients and 2 healthy controls, with a total of 58 ROIs, which may affect the generalizability of the conclusions. Additionally, there is no independent test set to validate these findings. The validation relied on statistical significance tests (proportion test, NCEM analysis), but no independent dataset was used for external validation.

Overall, TrimNN introduces a novel framework for identifying spatial motifs in omics data, demonstrating potential in biomarker discovery and disease characterization. However, its reliance on local topology may overlook long-range interactions, and the last two case studies are based on small datasets without independent validation, limiting generalizability. Further validation on larger, independent datasets and methodological improvements to capture broader spatial dependencies would enhance its robustness and applicability. Given these limitations, a major revision is necessary to ensure the robustness and applicability of the proposed method.

(Remarks on code availability)

I run the demo code and dataset, they work well.

Reviewer #3

(Remarks to the Author)

(Remarks on code availability)

Version 1:

Reviewer comments:

Reviewer #1

(Remarks to the Author)

The authors have addressed all my comments and the manuscript is much improved as a result. I have no further concerns.

(Remarks on code availability)

Well organised and documented code.

Reviewer #2

(Remarks to the Author)

The authors have satisfactorily addressed all of my previous concerns. The manuscript has been improved accordingly, and I recommend it for publication.

(Remarks on code availability)

Reviewer #3

(Remarks to the Author)

(Remarks on code availability)

Reviewer #1

This manuscript presents TrimNN, a graph-based deep learning framework that identifies cellular community (CC) motifs from spatial omics data, offering an interpretable, bottom-up approach to analysing multicellular topological organisation. The method demonstrates computational efficiency and biological relevance in Alzheimer's disease and colorectal carcinoma datasets, and has a broad applicability to a variety of spatial datasets. However, while the study appears methodologically sound and potentially impactful, several concerns should be addressed before publication, particularly regarding biological validation, methodological clarity and benchmarking.

Response: We thank the reviewer for the positive comments on our approach and greatly appreciate all the constructive suggestions and critiques, which have helped us further improve the quality of our manuscript. We added several extensions and evaluations and included more details regarding biological validation, methodological clarity, and benchmarking.

1. The idea of describing cellular communities in the form of graph motifs with variable size is powerful and well suited to spatial data. The authors state that their method's computational efficiency allow them to scale up to considerably large motifs. However, in all their analyses they only present insights from size 1-4 motifs, i.e. communities comprising only up to 4 cells. As such, these insights appear rather limited, even if some of them have biological relevance. What about larger motifs? Are they not relevant/informative? What motif size is most biologically meaningful? It would be very interesting if the authors could devise a way to scan through motifs of various sizes and determine the most suitable configuration of CC motifs (of varying sizes) that best describes the observed data in the slides.

Response: Thanks for your insightful observation and suggestions. According to the reviewer's comments, we expanded the identified size-3 motifs to subsets of size-12 in the context of AD research, extended the robustness simulation to size-5 (details in response to **Reviewer 1, Comment 2**), and performed a demo simulation of size-7 (details in response to **Reviewer 2, Comment 3**).

1.1 Challenges from statistical rigor.

Investigating large-size motifs is a computationally expensive task, and the field currently lacks rigorous approaches to perform statistical and biological evaluations. From the perspective of statistics, limited approaches are available to generalize statistical significance comparison across motifs of different sizes. We used Fisher's exact test (or the Chi-squared test) with odds ratios as the effect size within size-2 and size-3. However, further statistical evaluation in sizes larger than 3 becomes highly complex. Specifically, although a three-way contingency table can be analyzed using the Cochran–Mantel–Haenszel (CMH) test (PMID: 13655060) with the effect size measured by the Mantel–Haenszel common odds ratio, there is no single statistical test that is comparable to Fisher's exact test (or the Chi-squared test) or the CMH test—that universally applies to four-way or higher-dimensional contingency tables. Furthermore, analyzing three-way or higher-order

tables involves more complex hypothesis testing, as conditional and marginal associations may differ substantially (Agresti, Alan. Categorical data analysis. John Wiley & Sons, 2013.).

1.2 Our solution.

In the revision, we borrowed the idea from functional Gene Ontology (GO) and pathway enrichments to perform the significance evaluation of differentially expressed genes (DEGs) from different sizes of CC motifs. This evaluation can indirectly provide a “fair evaluation” of which size of CC motif best describes the biology from the observed data in the slides, as these CC motifs are hypothetically associated with AD biological and pathological mechanisms.

Given the exponential computational complexity associated with large motifs, we investigated their biological relevance by identifying biologically meaningful motifs in the AD study containing only ‘CTX-Ex’ (cell type C) and ‘Microglia’ (cell type M). In the assessment, we extended our motif size up to 10 triangles (typically size 12) (shown in **Supplementary Fig. 37**). We benchmarked the exact occurrences of each motif using the enumerative searching approach VF2 with guaranteed accuracy. We expand the strategy of gene-level interpretation in the main text of the AD case study: For each motif in multiple sizes, we identified its unique DEGs by comparing cells within the motif regions, as identified by VF2, to those outside the regions. GO and pathway enrichment analyses were then performed on these motif-related DEGs. The quality of CC motifs was evaluated and compared by adjusted p-values of enriched GO terms and pathways. To establish a baseline for AD relevance, we selected the top 300 GO terms found in the intersection of significant terms from four size-3 motifs: 'CCC', 'CCM', 'CMM', and 'MMM' in 13-month-old AD disease samples.

For each motif, Cauchy meta-analysis (PMID: 33012899) was applied to combine the p-values of these selected GO terms, with the resulting $-\log(p\text{-value})$ as a quantitative measurement of relevance to AD (**Supplementary Data 69**). **Supplementary Fig. 38 ABC** illustrates CC motifs' AD relevance using enriched top 300 GO terms on DEGs. Each dot in the plots represents a specific CC motif at the given motif size (X-axis), with the corresponding Y-axis value indicating the quantitative AD-relevance of that CC motif to the enriched top 300 GO terms. This quantitative relevance represents each motif as a dot in the plots. These motifs vary from size-1 pure cell type to triangle-10 (size-12) in homogeneous CTX-Ex (A), homogeneous Microglia (B), and heterogeneous combinations (C). To prevent potential bias, we also performed a similar analysis using all 821 significant GO terms (**Supplementary Data 70**) intersection of the four mentioned size-3 motifs (**Supplementary Fig. 38 DEF**). In both cases, we observed non-increasing trends in significance with increasing motif sizes. Motifs in smaller sizes, usually not exceeding 4, exhibited the most significant enriched results. Notably, the largest homogeneous microglia motifs were found at size 7, consistent with the observation that microglia are scattered and aggregated, unlike widely spread CTX-Ex cells.

We also found some large-size motifs that have better results on specific GO terms. Some motifs are equally significant across all sizes in ‘synapse assembly’ (**Supplementary Fig. 38 GHI**), and homogeneous CTX-EX motifs in size 6-7 have the most significant results in the GO term ‘positive regulation of glial cell proliferation’ (**Supplementary Fig. 38 JKL**). In pathway enrichment analysis, we observed a similar non-increasing trend in all 27 significant pathways intersecting the four mentioned size-3 motifs in the main text (**Supplementary Data 71** and **Supplementary Fig.**

39 ABC). This trend does not hold true when going to specific pathways. Similarly, we observe the plateau of significance on ‘Neuronal System’ (**Supplementary Fig. 39 DEF**) and the descending-ascending-descending trend in the Pathway ‘Axon guidance’ (**Supplementary Fig. 39 GHI**).

In addition, we performed two additional experiments: extended the robustness simulation to size-5 to show the robustness of multi-size motifs resistant to noises (details in response to **Reviewer 1, Comment 2**), and a specially designed system of size-7 to demo the idealized process in identifying the appropriate size of CC motifs (details in response to **Reviewer 2, Comment 3**).

1.3 Summary.

In summary, even large-size motifs reveal their values in top-down approaches, the theory and practice of finding the most appropriate size to describe the observed data in spatial biology is still in its infancy. From the limited experiments and observations, different tissues in different studies within different technologies might have different optimal sizes of motifs representing underlying biology. Different approaches in diverse statistics, functional analysis, and phenotypic analysis could reveal distinctive perspectives in different biological scenarios. Notably, it is even very difficult to generate these large-size motifs for evaluation in practice. These computationally extensive experiments exceed the capacity of our HPC of 128 CPU cores within weeks of running time. We believe TrimNN provides an accurate and computationally feasible way to approach this challenge.

We put these investigations into the sections of Discussion and Methods, and listed them as one of our further directions in Lines 654-666 and Lines 966-977.

Supplementary Fig. 37 Topology of CC motifs composed of triangles from Triangle-1 to Triangle-10. The total motif number denotes the number of unique CC motifs within two cell types, such as 'CTX-Ex' and 'Microglia' as node types in the applied Alzheimer's disease research.

Supplementary Fig. 38 Assessment of biological relevance using GO enrichment analysis of DEGs within and out of multiple-size CC motifs. The Y-axis represents the enrichment of top 300 AD-related GO terms of DEGs from CC motifs in Homogenous CTX-Ex (**A**), Homogenous Microglia (**B**), and Heterogenous CTX-Ex and Microglia (**C**). Enrichment on all 821 AD-related GO terms of DEGs from motifs in Homogenous CTX-Ex (**D**), Homogenous Microglia (**E**), and Heterogenous CTX-Ex and Microglia (**F**). Enrichment on GO term ‘Synapse Assembly’ of DEGs from motifs in Homogenous CTX-Ex (**G**), Homogenous Microglia (**H**), and Heterogenous CTX-Ex and Microglia (**I**). Enrichment on GO term ‘Positive Regulation of Glial Cell Proliferation’ of DEGs from motifs in Homogenous CTX-Ex (**J**), Homogenous Microglia (**K**), and Heterogenous CTX-Ex and Microglia (**L**). The X-axis represents the motifs increasing in size. Each data point represents the $-\log(p\text{-value})$ of enrichment on specific GO terms from DEGs within and out of motif regions. Here AD-related GO terms represent the

intersected significant GO terms among size-3 motifs – ‘CCC’, ‘CCM’, ‘CMM’, and ‘MMM’ in 13-month-old disease samples in AD research.

Supplementary Fig. 39 Assessment of biological relevance using KEGG Pathway enrichment analysis of DEGs within and out of multiple size CC motifs. The Y-axis represents the enrichment of all 27 AD-related pathways of DEGs from CC motifs in Homogenous CTX-Ex (A), Homogenous Microglia (B), and Heterogenous CTX-Ex and Microglia(C). Enrichment on Pathway ‘Neuronal System’ of DEGs from motifs in Homogenous CTX-Ex (D), Homogenous Microglia (E), and Heterogenous CTX-Ex and Microglia (F). Enrichment on Pathway ‘Axon guidance’ of DEGs from motifs in Homogenous CTX-Ex (G), Homogenous Microglia (H), and Heterogenous CTX-Ex and Microglia (I). The X-axis represents the motifs increasing in size. Each data point represents the $-\log(p \text{ value})$ of enrichment on specific Pathways from DEGs within and out of motif regions. Here AD-related Pathways represent the intersected significant Pathways among size-3 motifs – ‘CCC’, ‘CCM’, ‘CMM’, and ‘MMM’ in 13-month-old disease samples in AD research.

- While I appreciate the question above is a rather difficult task, I would at least be curious to know how the performance metrics of TrimNn change in figures 2 a-c when adding noise if motifs of size >3 are investigated. It appears there may be a limit to the ability to

robustly capture these patterns, with the performance decreasing with the increase in the size of the motif and the amount of noise. I find this rather counterintuitive though – could the authors explain why the correlation is worse when introducing noise in larger motif structures?

Response: Thanks for your suggestion. Per the reviewer's request, the motif size in our simulation experiments has been extended to 5 to evaluate the method's robustness against heightened structural complexity rigorously. We also included results with noise levels of 0.3 and 0.4 to perform comprehensive simulations systematically. The complete results are updated in **Fig. 2 ABC** and the file Source Data.

For each motif size, we calculated the Spearman correlation between the abundance rankings of all possible motifs in the original and the noise-poised data. Notably, in cases of cell missing, if a particular motif is absent, we assign it an abundance of zero and include it in the ranking accordingly. If the simulated noise does not change the relative abundance, the CC motifs derived by the adopted Delaunay Triangulation are robust to noises, and the correlation will be high. We observed that correlation decreases linearly across all three simulation types with increasing noise and motif size. We want to mention that the lengths of rank lists of all the possible size-1, size-2, size-3, size-4, and size-5 motifs are 13, 91, 455, 8281, and 54,418, respectively. The probability of occurrences of size-1 motifs is $1/13$ and $1/54,418$ in size-5. Larger motifs yield more enumerated patterns, making them less likely to maintain their status in the spatial spaces. Within the same noise level, large-size motifs are relatively sensitive to noises and easily change to another motif under the influence. This may explain the phenomenon that correlation is worse in larger motif structures.

Within the updated simulations up to size-5, we agree with the reviewer's observation that the robustness of the motifs indeed has limitations. In both scenarios of missing cell (**Fig. 2A**) and shifting cell coordinate (**Fig. 2B**), the correlation values remain relatively stable even under extreme noise conditions (noise in half cells with level 0.5). For cell type misclassification (**Fig. 2C**), the correlation values deteriorate for large motif sizes when the noise level is high. However, such extremely high noise levels are unlikely to occur in practical scenarios in cell type annotation. Therefore, our approach is expected to be robust in practical applications.

Similar to **Question 1**, we are at our current HPC capacity of 128 CPU cores to generate systematic results in size-5 with exponential complexity. Within the linear tendencies illustrated from size-1 to size-5, we anticipate the performances in correlation getting worse with increasing motif sizes. It should not influence our current results in size-3 and size-4 CC motifs, but it is an open question for further exploration in the relevant research community.

These revisions and descriptions have been incorporated into Lines 151-155 and **Fig. 2**.

Fig. 2 The performance of TrimNN on spatial omics. Simulations of cell missing effects (A), cell coordinate shifting effects (B), and cell-type misclassification effects (C) on CC motifs, represented as Spearman correlation between abundance rankings of all the possible motifs before and after simulated noises at cell proportions of 0.01, 0.05, 0.1, 0.2, 0.3, 0.4, and 0.5 within CC motifs in size-1, size-2, size-3, size-4, and size-5.

3. It is not clear how the synthetic training datasets were generated. Specifically:

- How were node type permutations constructed? Were there any constraints to ensure biological plausibility?

Response: Thanks for allowing us to clarify the confusion. In constructing the training data, we deliberately avoid any prior knowledge-based biases. The problem has already been formulated as a mathematical problem, and the node types were randomly assigned with one of the cell types in the cellular community under a uniform distribution. There are no biological constraints here and this fully data-driven strategy intentionally excludes biological assumptions, as the limited availability of validated domain knowledge precludes the adoption of parameterized distributions (e.g., Negative Binomial) that inherently encode theoretical presuppositions. The reproducibility codes are located in the Figshare depository, and an updated description is provided in the Method section in Lines 802-803.

- Given that VF2 was used for ground truth, could this introduce biases, particularly in how TrimNN learns to identify motifs?

Response: We believe adopting the VF2 algorithm as the ground truth reference does not introduce systematic biases, as it operates through a well-established mathematical framework of depth-first searching (DFS, backtracking) for exact subgraph isomorphism counting. Designed on general graphs, this method strictly adheres to its canonical subgraph isomorphism operations without incorporating external domain-specific data or modeling assumptions, thereby preventing the introduction of confounding variables or theoretical presuppositions during occurrence frequency quantification. Many established graph processing packages, such as igraph

(<https://igraph.org/c/doc/igraph-Isomorphism.html>) and Networkx (<https://networkx.org/documentation/stable/reference/algorithms/isomorphism.vf2.html>), implement VF2 for diverse graph-related tasks.

- Were real-world spatial omics datasets incorporated into the training process, or was it based entirely on synthetic data? If synthetic, how well does it generalise to experimental datasets?

Response: Our training process is entirely based on synthetic data to rigorously preserve the generalized graph-theoretic formulation of our overrepresented motif identification task. By abstracting the problem into a domain-agnostic graph pattern recognition framework, we intentionally circumvent biases inherent in real-world spatial omics datasets (e.g., platform-specific noise profiles, batch effects, or tissue preparation artifacts). We think this approach ensures that the model's learning dynamics are driven solely by topological pattern characteristics. As all synthetic and experimental datasets applied the same processing steps to get cellular communities both entirely composed of triangles, we think these synthetic results can be, at least partially, generalized to experimental datasets.

4. Some biological interpretations remain unclear:

- In the Alzheimer's disease dataset, how do the identified motifs compare with previously known functional tissue units (FTUs) or clustering-based spatial patterns? Are these motifs confirming existing biological knowledge, or do they provide new insights?

Response: Thank you for giving us an opportunity to clarify our findings.

Functional Tissue Units (FTUs) are defined as the smallest tissue organization that performs a unique physiologic function and is over-represented multiple times in a whole organ (PMID: 39934102). We found that our defined CC motifs are very connected but not necessarily FTUs. Currently, the largest effort in curating human FTUs is the Human Reference Atlas (<https://humanatlas.io/>), which recruits 26 consortia with 250+ experts but only lists 22 FTUs in 10 organs in their recent data release 2.2. In this case study of Alzheimer's disease, it looks like the concept of FTUs is still evolving. We would like to be cautious about claiming the connections between our findings and known FTUs, but we believe there will be progress in this direction.

CC motifs share some common principals with clustering-based spatial patterns but differ in the following aspects (also shown in **Supplementary Fig. 1**): (1) CC motifs are strictly defined as cell type topological combination, independent of the specific coordinate values, expression matrix, or geometric structure of the input. But spatial patterns are usually ad-hoc defined by clustering specific input data and may differ in other inputs. (2) CC motifs usually have smaller cell sizes due to computational constraints, but clustering-based spatial patterns can have more cells. (3) CC motifs, especially in large sizes, are usually not easy to get because they grow exponentially with increasing sizes, but getting spatial patterns is relatively faster with the help of clustering algorithms. (4) CC motifs can be more interpretable on multiple levels, including statistics, DEG, cell-cell communication, and phenotypical discrimination. However, spatial patterns from clustering are usually interpreted only at the cell composition level. In many ways, the relationship between clustering-based approaches and CC motifs is similar between functional domains (such

as Pfam (PMID: 22127870)) and sequence motifs (such as PROSITE (PMID: 19858104)) in protein research. We also put this discussion in the Discussion section.

Following the reviewer's request, we compared our results with results from clustering-based methods Banksy and SPACE-GM using the same dataset in the AD case study. In both methods, we selected 13, the number of cell types, as the number of clusters. Banksy identified 13 microenvironments within each of the 8 available samples. The hierarchical clustering method clustermap identified 7 clusters across these 8 samples based on cell type compositions (**Supplementary Fig. 33**). We can see that almost every identified cluster has an even proportion of disease and control samples in the upper right corner of **Supplementary Fig. 33**.

Similarly, SPACE-GM obtained 13 microenvironments as clusters on learned embeddings (**Supplementary Fig. 34A**), and we computed the occurrence counts of the identified microenvironments in each sample. All the steps followed official tutorials. However, none of the 13 microenvironments showed significant differences in student t-tests between control and disease groups (**Supplementary Fig. 34B**). There may be extra room to improve with sophisticated optimization in parameter selection, but both results from clustering-based Banksy and SPACE-GM show they might not be directly effective in capturing the spatial patterns features to distinguish disease and control samples.

In contrast, CC motifs composed of cell type CTX-Ex and Microglia identified by TrimNN show significant differences between disease and control samples, details in the section of 'TrimNN identifies CC motifs revealing diverse roles in Alzheimer's disease using spatial transcriptomics data' and **Fig. 4**. In AD dataset used in the research, we assume these CC motifs provide a more sensitive approach in identifying differential spatial patterns than clustering results from top-down approach. We rephrased the corresponding paragraphs in the section.

In concept, we think these identified CC motifs can represent multi-level biological knowledge in terms of cell type, gene, and cell-cell communications. (i) At the cell type level, these motifs compose cell types in CTX-Ex (cortex excitatory neurons) and Microglia, which are known to play important roles in the pathogenesis of Alzheimer's disease. (ii) At the gene level, functional GO and pathway enrichment analysis confirm that DEGs between cells within and out of the motifs align with known functional roles in AD. Some of the most significant GO terms including 'regulation of synapse structure or activity', 'learning or memory', and 'synapse assembly', and enriched Pathways including 'Neural System', 'Axon guidance', and 'Transmission across Chemical Synapses', aligning with the knowledge of AD. (iii) At the cell-cell communication level, CellChat and DeepTalk reveal significantly more interactions originating from motif-enriched regions than complementary regions. Notably, interactions in non-motif regions, such as pathways like 'NRG', 'PDGF', 'SEMA3', and 'VEGF', were less prominent.

We also would like to claim that these identified CC motifs enabled new insights into spatial biology: As an unbiased data-driven approach, TrimNN identified two distinct types of CC motifs corresponding to different mechanisms in complex AD pathology. Given knowledge of Microglia's contribution to neuroinflammation and amyloid- β toxic to neurons, CC motif analysis nascent the spatial roles of CTX-Ex and Microglia by decomposing the spectrum of amyloid- β spatial co-localizations within scientific rigor. Based on the spatial relations to the AD hall marker

amyloid- β , we independently found that the extent of homogeneity of microglia regions seemed to correspond to a larger co-occurrence probability of amyloid- β . In contrast, the extent of homogeneity of cortex excitatory neurons tallied to lower co-occurrence probability. This trend prevailed across the whole spectrum of CC motifs composed of microglia and cortex excitatory neurons in multiple sizes, from a very high ratio of size-4 ‘MMM’ to a very low ratio of size-3 ‘CCC’, which makes a continuous and consistent spectrum between these cell types (Fig. 4L). More biological analyses, such as cell-cell communications (Fig. 4EFGH) and expression-based hierarchical clustering (Fig. 4K), also support the distinct roles of these motifs. Of course, these findings are still hypothetical and need in-depth biological validations in the future.

We updated the Discussion in Lines 590-616, and the original paragraphs have been reorganized in Lines 444-454.

Supplementary Fig. 33 Hierarchical clustering on all Banksy-defined microenvironments on each of 8 Alzheimer’s disease samples. Each sample identified 13 microenvironments. The Y-axis represents 7 clusters shared in all microenvironments of AD samples. The X-axis represents cell type composition. The top right corner shows the proportion bar plot of disease and control samples in each cluster. The number in the parentheses shows the number of samples in the disease and control groups. These clusters do not directly differentiate AD and non-AD samples.

Supplementary Fig. 34 A left: Hierarchical clustering on embeddings learned from SPACE-GM on Alzheimer’s disease samples. Each sample identified 13 microenvironments. The Y-axis represents

13 identified microenvironments. The X-axis represents cell type composition. **B.** Jitter plots of the number of microenvironments in AD control and disease samples. None of the plots shows significance in the student t-test, demonstrating these clusters do not directly differentiate AD and non-AD samples.

- The colorectal cancer survival analysis (Figure 3) is promising, but it requires validation in an independent dataset to confirm its robustness. Have the authors explored external datasets or cross-validation strategies?

Response: We made considerable efforts to identify independent datasets for further validation. Unfortunately, we were unable to find additional publicly available colorectal cancer (CRC) datasets that include both spatial omics and survival information until May 1st, 2025. The most challenging part is publicly available survival information, which is rarely included in current spatial omics studies. Instead, we validated the key results indirectly on size-1 motifs as cell types using the CRC-related COAD and READ cohort of The Cancer Genome Atlas (TCGA). We selected macrophage marker genes: CD68, CD163, CD14, ITGAM (PMID: 23619691), and smooth muscle marker genes ACTA2, MYH11, and MYL9 (PMID: 24419809) for this validation.

Based on these marker gene expressions, we plotted survival curves in **Supplementary Figs. 8-9** using a web portal at <http://www.tcg-survival.com> (PMID: 35354049). As shown in the figures, the survival curves of the size-1 CC motifs (macrophage and smooth muscle) represented by these marker genes do not exhibit significant separation in survival (Cox PH >0.05) between the low and high expression groups. In comparison with size-2 to size-4 motifs in **Fig. 3D–F**, these independent results from TCGA partially support the observation in this case study that pure cell type composition of macrophage and smooth muscle cannot distinguish survival. As the motif size increases, CC motifs become more effective in differentiating patient survival.

These revisions and descriptions have been incorporated into Lines 306-313.

Supplementary Fig. 8 Survival curves associated with CD68⁺CD163⁺ Macrophage in TCGA. Survival Curves of high vs. low expression of CD68⁺CD163⁺ Macrophage marker genes: **A.** CD68, **B.** CD163, **C.** CD14, **D.** ITGAM in TCGA-COAD. Survival Curves of high vs. low expression of CD68⁺CD163⁺ Macrophage marker genes: **E.** CD68, **F.** CD163, **G.** CD14, **H.** ITGAM in TCGA-READ.

Supplementary Fig. 9 Survival curves associated with smooth muscle in TCGA. Survival curves of high vs. low expression of smooth muscle marker genes: **A**. ACTA2, **B**. MYH11, and **C**. MYL9 in TCGA-COAD. Survival curves of high vs. low expression of smooth muscle marker genes: **D**. ACTA2, **E**. MYH11, and **F**. MYL9 in TCGA-READ.

- In the colorectal carcinoma study presented in Figure 5, I find the changes in CC patterns less biologically interpretable. What are the functional implications of the shifted vs homeostatic interaction motifs? Could these motifs be linked to immune evasion, tumour progression, or response to therapy? If so, additional discussion would help contextualise their significance.

Response: We appreciate this comment and have added new interpretations to clarify the biological meaning of these motif patterns. The Shifted Interaction Motif ('ABC' to 'ABCD') shows increased expression of immune-related and metabolic proteins in disease samples, suggesting more immune involvement but also possible immune dysfunction and tumor progression. For example, CD39 is upregulated and may reflect immunosuppressive activity, while markers like

Ki67 and HK1 suggest increased proliferation and metabolism. The Homeostatic Interaction Motif ('AEC' to 'AECC') remains stable in frequency, but shows higher PD1 and GLUT1 levels in disease, pointing to immune checkpoint activation and metabolic stress. These patterns help explain how spatial motif changes relate to immune evasion and tumor behavior, and may have implications for therapy, including response to checkpoint inhibitors or metabolic-targeting strategies.

These revisions and descriptions have been incorporated into Lines 522-553.

5. The manuscript discusses the limitations of clustering-based approaches but lacks a quantitative performance comparison with other spatial omics tools. To strengthen the argument for TrimNN's advantages, the authors should provide a direct performance comparison tools such as CytoCommunity, SPACE-GM, and BANKSY on the same datasets, and benchmark TrimNN's biological motif discovery against standard clustering approaches to highlight what novel insights are uniquely captured.

Response: Thank you for your valuable suggestions that significantly improve our manuscript.

While clustering-based approaches such as SPACE-GM and CytoCommunity primarily focus on direct phenotype classification. TrimNN emphasizes the identification of differential CC motifs. These motifs, though not derived from direct phenotype supervision, capture meaningful topological patterns that are highly relevant to disease states. Therefore, we can use it as a feature to help a simple model achieve effective clustering. As a result, TrimNN offers complementary insights that not only enhance the interpretability of cellular interactions but also contribute effectively to phenotype discrimination from a structural and mechanistic perspective. We provided a comprehensive comparison between this method with clustering-based approaches in CytoCommunity and SPACE-GM, and expanded our evaluation metrics to include the widely-used criteria, including accuracy, precision, recall, F1-score, MCC, Precision-AUC, and ROC-AUC provided by Python's *scikit-learn* package. We did not include BANKSY in this comparison, as the tool is specifically designed for cell type annotation and segmentation, and its results as clusters cannot directly be compared with phenotypical labels in the formulation of classification. The performance of the logistic regression model using the top 29 TrimNN motifs, CytoCommunity, and SPACE-GM is detailed in **Supplementary Data 11**. We observe that the logistic regression model with TrimNN's top 29 motifs outperforms CytoCommunity and SPACE-GM in almost all the quantitative metrics except Precision-AUC, where CytoCommunity's result (0.833 vs. 0.760). One of the most important criteria is ROC-AUC, where logistic regression's 0.796 leads to CytoCommunity's 0.714 and SPACE-GM's 0.765. Although the authors did not share the code of CytoCommunity for this specific CRC task, we carefully followed the tutorials they provided, and adhered to the recommended parameter settings. All the analysis codes are deposited in the Figshare repository for reproducibility.

We would like to highlight the uniqueness of the CC motifs identified by TrimNN compared with clustering-based methods in the Discussion section. (1) *It overcomes the limitation of clustering approaches.* Without arbitrary parameters in clustering, the bottom-up approach identifies topological building blocks as countable CC motifs to represent CNs (**Fig. 3A, 3B**). (2) *It provides intrinsic interpretability of the topological building blocks of CNs.* Easily interpretable biologically

at the cellular (Fig. 4E, 4F, 4G, and 4H) and gene levels (Fig. 4I, 4J, 4K, and 5J), and pathologically intertwined with clinical phenotypes (Fig. 3D, 3E, 3F), the results of TrimNN demonstrate the colocalization differences between repeated motifs and disease markers (Fig. 4L), which may correspond to different biological and pathological hypotheses. (3) *It ensures better generalizability across samples.* Without a complex machine learning model with uninterpretable embeddings dependent on trained samples, CC motifs as explicit interpretable representations are robust to differentiate CNs (Fig. 3C) and noises (Fig. 2A, 2B, and 2C). (4) *It provides new biological hypotheses in complex diseases.* In the CRC study, enrichments of CC motifs in multiple sizes are shown to be associated with survival. In the AD case study, we show a spectrum of spatial co-localization between the homogeneity of specific cell types and AD hallmarks. In the colorectal carcinoma study, two types of CC motifs are inferred to have different roles in topological combinations within incremental sizes. Clustering-based methods do not deliver these results uniquely captured by the proposed approach.

These revisions and descriptions have been incorporated into Lines 261-263 and the section of Discussion at Lines 590-601.

6. While the discussion highlights TrimNN's advantages, it does not fully address potential methodological biases:

- Delaunay triangulation is used to define CCs, but could this introduce artefacts in tissues with irregular or non-uniform spatial distributions?

Response: Thank you for reminding us to explore the question. Yes, we found that Delaunay triangulation can exhibit irregular or non-uniform spatial distributions in certain boundary regions. To address this issue, we added preprocessing steps for noise reduction with unnecessary edges. We computed the lengths of all edges and used the 99th percentile as a threshold to remove outlier edges that are too long. Users can customize this preprocessing step as an optional step in our updated TrimNN package in the GitHub repository.

These descriptions have been incorporated into Lines 839-844.

- The semi-greedy pattern growth strategy is an interesting choice, but alternatively, could reinforcement learning approaches further refine the search process?

Response: Yes, we believe that reinforcement learning can play a more effective role in pattern growth, but it is not a trivial implementation. We will explore this strategy in our next steps.

7. The authors use DESeq2 for DEGs analysis between motif and non-motif regions. Since DESeq2 is specifically designed for bulk (negative binomial GLM to handle overdispersion) and requires raw counts (which is quite high in bulk data), I'm not sure if this is the best method to be applied in this instance.

Response: We appreciate your valuable suggestion and have revised the DEG analysis as suggested. DESeq2 is specifically designed for bulk RNA-seq due to its ability to handle overdispersion with a negative binomial GLM, where the number of reads per gene is relatively high and stable. But this does not hold for single-cell data, which tends to have sparse, low-count

gene expression profiles. We agree that DESeq2 might not be the optimal choice to fully capture the complexities of scRNA-seq data.

In the revision, we used the Wilcoxon rank-sum test for the DEG analysis instead of DESeq2. Adopted by the popular tool Seurat, the Wilcoxon rank-sum test is widely used in scRNA-seq analyses. Unlike DESeq2, with the assumption of a negative binomial distribution, the Wilcoxon rank-sum test uses a non-parametric test to handle the irregularities and sparsity inherent in single-cell datasets. Therefore, to better fit the single-cell scenario and ensure more reliable results, our study adopted the Wilcoxon rank-sum test for DEG analysis.

For scientific rigor, we compared the obtained DEGs using both the Wilcoxon rank-sum test and DESeq2 with the same parameter settings in 13-month-old AD samples in the AD case study. We can see there are some differences between these two methods. We also performed GO and pathway enrichment analysis on the DEGs from both methods and included the comparison in the **Extended Fig. 1** below.

Extended Fig. 1 Comparison of GO enrichment analysis of Biological Processes using DEGs from Wilcoxon rank-sum test and DESeq2 in motif ‘CCC’ **A.** and **B.**, motif ‘CCM’ **C.** and **D.**, motif ‘CMM’ **E.** and **F.**, and motif ‘MMM’ **G.** and **H.** in thirteen-month-old AD samples. Comparison of Pathway enrichment analysis using DEGs from Wilcoxon rank-sum test and DESeq2 in motif ‘CCC’ **I.** and **J.**, motif ‘CCM’ **K.** and **L.**, motif ‘CMM’ **M.** and **N.**, and motif ‘MMM’ **O.** and **P.** in thirteen-month-old AD samples.

We updated **Fig. 4. I J** and the corresponding **Supplementary Figs. 28-29** with the results from the Wilcoxon rank-sum test and made the necessary revisions in the manuscript at Lines 388-400 accordingly.

Fig. 4 I. GO enrichment analysis of Biological Processes and **J.** Pathway enrichment analysis on DEGs between regions containing and not containing the ‘MMM’ motif in thirteen-month-old AD samples.

Minor point: The manuscript could benefit from proofreading and support by a native English speaker to make the text easier to read and follow and correct some grammatical errors.

Response: Thank you for your suggestion. We have used proofreading services from a professional company located in the US to improve the manuscript's readability.

(Remarks on code availability)

I have not tested the code myself, but have reviewed the github repository and it is very well organised and documented. It appears very straightforward to run, with a very clear input that is easily generated from spatial omics data.

Response: We thank the reviewer for the positive comment.

Reviewer #2

Yu & Wang et al. developed a cellular community (CC) motif discovery tool named TrimNN, designed to identify and quantify the abundance of k-mer CC motifs in spatial omics (ST) data. By leveraging a graph-based deep learning approach, TrimNN enables the detection of conserved multicellular spatial patterns across different biological contexts. The authors validate TrimNN through three case studies: (1) Colorectal cancer (CRC), where TrimNN successfully distinguishes patient subtypes (Crohn's-like lymphoid reaction (CLR) vs. diffuse inflammatory infiltration (DII)) based on CC motif abundance, revealing potential prognostic biomarkers. (2) Alzheimer's disease (AD), where CC motifs associated with microglia and excitatory neurons are linked to β -amyloid deposition and disease progression. (3) Colorectal carcinoma tumor microenvironment (TME), where TrimNN identifies distinct immune-epithelial interaction motifs, highlighting changes in immune cell spatial organization between healthy and tumor tissues. The study presents an innovative computational framework with potential applications in spatial biology, biomarker discovery, and disease characterization. However, several critical aspects require further clarification and validation, as outlined below:

Response: We thank the reviewer for the positive comments and value the following constructive suggestions. Please find our point-by-point responses below.

1. One major concern is that the entire method relies on the topological connections of locally neighboring cells, potentially overlooking long-range cellular communication. This raises the question of whether the identified k-mer CC motifs primarily reflect basic spatial arrangements of cellular composition rather than capturing higher-order functional organization. For instance, the authors construct 4-mer motifs by growing patterns from 3-mer motifs, assuming that the local expansion sufficiently captures biologically meaningful motifs. However, long-range cellular interactions (or long-distance communication) could play a crucial role in tissue organization and

functon(see: <https://www.nature.com/articles/s41467-024-51329-2>), which this method might fail to capture. To address this limitation, the authors should provide strong biological evidence to support the assumption that long-range interactions are not significantly influential in defining meaningful CC motifs. Alternatively, they should discuss potential methodological extensions that incorporate long-range interactions, such as weighted cell-cell interaction graphs based on cell communication networks (e.g., CellChat), or incorporating graph attention mechanisms to learn distant dependencies (e.g., DeepTalk).

Response: Thank you for your suggestion regarding the consideration of long-range cellular interactions. We would like to clarify that our work is not designed to capture cell-cell communications. In this work, we mainly use cell-cell communications to validate our results in representing cell organizations, and there may be a huge gap inferring cell organizations directly from cell-cell communications. Even though we agree that long-range cellular interactions are crucial in many biological processes, the forces driving cell organization to perform specific functionalities are another underexplored topic, where long-range cellular interactions may play roles. We will continue exploring this direction in the future.

Based on the reviewer's recommendation, we extended experiments using DeepTalk on our Alzheimer's disease case study with spatial transcriptomics data besides CellChat. Following the official tutorial of DeepTalk, we first integrated STARmap PLUS AD dataset and scRNA-seq data GSE176032 downloaded from Gene Expression Omnibus (GEO). We performed DeepTalk considering long-range cellular communications on the integrated data set and focused on the pathway-related ligand-receptor pairs identified by CellChat in **Fig. 4E–H**.

These results demonstrate that DeepTalk independently confirmed long-range cellular interactions show significant differences between motif and non-motif regions within AD-related ligand-receptor pairs, which supports the results identified from TrimNN. As shown in **Supplementary Fig. 24B and H**, we observed significant differences in communication scores between cell types of cortex and microglia within the ligand-receptor pairs Csf1-Csf1r (Mann-Whitney U-test p-value: 0.0326) and Vegfa-Flt1 (Mann-Whitney U-test p-value: 0.0029) between Cortex-Cortex-Cortex (CCC) and non-CCC regions. These two ligand-receptor pairs belong to two AD-related pathways: CSF and VEGF, respectively. The DeepTalk results differentiating CCC, non-CCC, Microglia-Microglia-Microglia (MMM), and non-MMM regions between cell-cell communications from cortex excitatory neuron to microglia and microglia to cortex excitatory neuron are shown in **Supplementary Figs. 22-25**. Besides pair-wise comparison between cortex excitatory neurons and microglia, we also compared the heatmap of the communication scores between all cell types, which showed significant differences in all mediated ligand-receptor pairs (Mann-Whitney U-test p-value <0.001) (**Supplementary Figs. 26-27**).

We incorporated the related description in sections of Results and Methods to Lines 380-387, Lines 984-988 of the manuscript.

Supplementary Fig. 22 Comparison of Cortex (C)-to-Microglia communication scores between CCC (left) and non-CCC (right) regions mediated by different Ligand-Receptor (L-R) pairs using DeepTalk. The box plot shows the differences in L-R pair **A.** Ccl3-Ccr5, **B.** Csf1-Csf1r, **C.** Gdf15-Tgfr2, **D.** Nrg3-ErbB4, **E.** Pdgfa-Pdgfra, **F.** Pdgfa-Pdgfrb, **G.** Pyy-Npy1r, **H.** Vegfa-Flt1, and **I.** Vip-Vipr2. The Mann-Whitney U-test is used to test the significance level.

Supplementary Fig. 23 Comparison of Cortex-to-Microglia (M) communication scores between MMM (left) and non-MMM (right) regions mediated by different Ligand-Receptor (L-R) pairs using DeepTalk. The box plot shows the differences in L-R pair **A. Ccl3-Ccr5**, **B. Csf1-Csf1r**, **C. Gdf15-Tgfr2**, **D. Nrg3-Erb4**, **E. Pdgfa-Pdgfra**, **F. Pdgfa-Pdgfrb**, **G. Pyy-Npy1r**, **H. Vegfa-Flt1**, and **I. Vip-Vipr2**. The Mann-Whitney U-test is used to test the significance level.

Supplementary Fig. 24 Comparison of Microglia-to-Cortex (C) communication scores between CCC (left) and non-CCC (right) regions mediated by different Ligand-Receptor (L-R) pairs using DeepTalk. The box plot shows the differences in L-R pair **A**. **Ccl3-Ccr5**, **B**. **Csf1-Csf1r**, **C**. **Gdf15-Tgfr2**, **D**. **Nrg3-Erb4**, **E**. **Pdgfa-Pdgfra**, **F**. **Pdgfa-Pdgfrb**, **G**. **Pyy-Npy1r**, **H**. **Vegfa-Flt1**, and **I**. **Vip-Vipr2**. The Mann-Whitney U-test is used to test the significance level. L-R pairs in a yellow rectangle are statistically significant.

Supplementary Fig. 25 Comparison of Microglia (M)-to-Cortex communication scores between MMM (left) and non-MMM (right) regions mediated by different Ligand-Receptor (L-R) pairs using DeepTalk. The box plot shows the differences in L-R pair **A. Ccl3-Ccr5**, **B. Csf1-Csf1r**, **C. Gdf15-Tgfr2**, **D. Nrg3-ErbB4**, **E. Pdgfra-Pdgfra**, **F. Pdgfra-Pdgfrb**, **G. Pyy-Npy1r**, **H. Vegfa-Flt1**, and **I. Vip-Vipr2**. The Mann-Whitney U-test is used to test the significance level.

Supplementary Fig. 26 Heatmaps of the communication scores between all cell types between CCC (left) and non-CCC (right) regions mediated by different L-R pairs using DeepTalk. Heatmaps show the differences in L-R pair **A.** Ccl3-Ccr5, **B.** Csf1-Csf1r, **C.** Gdf15-Tgfr2, **D.** Nrg3-ErbB4, **E.** Pdgfa-Pdgfra, **F.** Pdgfa-Pdgfrb, **G.** Pyy-Npy1r, **H.** Vegfa-Flt1, and **I.** Vip-Vipr2. The Mann-Whitney U-test is used to test the significance level.

Supplementary Fig. 27 Heatmaps of the communication scores between all cell types between MMM (left) and non-MMM (right) regions mediated by different L-R pairs using DeepTalk. Heatmaps show the differences in L-R pair **A. Ccl3-Ccr5**, **B. Csf1-Csf1r**, **C. Gdf15-Tgfr2**, **D. Nrg3-Erb4**, **E. Pdgfa-Pdgfra**, **F. Pdgfa-Pdgfrb**, **G. Pyy-Npy1r**, **H. Vegfa-Flt1**, and **I. Vip-Vipr2**. The Mann-Whitney U-test is used to test the significance level.

2. In Figure 3A, the authors compared their approach with CytoCommunity (dim = 512) on a dataset of only 35 advanced colorectal cancer (CRC) patients (17 with CLR vs. 18 with DII). A major concern is the discrepancy in feature dimensionality: CytoCommunity used 512-dimensional features, which far exceeds the number of samples (35), raising concerns about overfitting and model instability. In contrast, TrimNN used at most 100 CC motifs (or even as few as 29 motifs), making the comparison potentially unfair. The authors should clarify why a 512-dimensional feature space was used in CytoCommunity, given the small sample size. Furthermore, they should discuss how TrimNN's lower-dimensional motif representation avoids overfitting, particularly in small datasets.

Response: Thank you for your valuable advice and insightful suggestions.

The tutorial of CytoCommunity defines embedding dimension 512 as the default hyperparameter for supervised tasks (https://github.com/huBioinfo/CytoCommunity/blob/main/Tutorial/Supervised/Step2_TCNLearning_Supervised.py). We are wondering this 512 is the results of dimension reduction from the number of genes in the spatial transcriptome in their original publication, and we agree to be cautious when we applying in this datasets with 35 spatial samples. To better preserve their original ideas, we usually benchmark methods using their default settings. That is how we adopted this dimensionality for graph classification on the CRC patient dataset.

We agree that more features than samples cause overfitting in machine learning. To assess the impact of lower-dimensional feature spaces and mitigate concerns about overfitting, we follow the reviewer's suggestion by testing CytoCommunity with the same dimensionality of 29 as Logistic Regression on TrimNN. The ROC-AUC score obtained with 29 dimensions in CytoCommunity was 0.69 (as **Supplementary Fig. 5**), which is lower than the 0.71 achieved with the 512-dimensional features. It looks like reducing the dimensionality may not improve the performance of CytoCommunity for this specific task, or their presentations need more samples to be stratified, which limits their power in this dataset.

We would like to clarify that CC motifs can be treated as representations of cellular neighborhoods. CC motifs themselves are not directly related to overfitting. We agree that overfitting may relate to the inappropriate application of machine learning approaches when the number of features exceeds the number of samples. In this work, we use logistic regression, one of the simplest machine learning methods, to demonstrate CC motifs as features in these models reserve most information from the input spatial omics data. Similar results using the Random Forest and Support Vector Machine were achieved, similar to those achieved by logistic regression, further supporting the representational power of CC motifs (**Supplementary Data 8-10**).

Per the reviewer's request, we conducted further investigations and clarification on the potential influence of overfitting in logistic regressions using CC motifs from TrimNN by (1) manipulating model dimensionality and (2) using 10 times 10-fold cross validation. The choice of 29 motifs was driven by the fact that the number of cell types in CRC is 29, meaning that we cannot extract more than 29 distinct motifs at size-1. We conducted experiments with size-3 motifs at varying

dimensionalities with a series of top 5, 10, 15, and 20 motifs (**Supplementary Fig. 5** and **Supplementary Data 7**), where the number of features is all less than the sample size of 35. All the results are generated by reproducible 10 times 10-fold cross validation. The resulting ROC-AUC scores were 0.75, 0.77, 0.79, and 0.78, respectively. We would like to mention that these cross-validated results with diverse small numbers of features are relatively stable. It looks like the model has little influence from overfitting with logistic regression, at least in this CRC dataset. The reproducibility codes are also included in the Figshare repository.

We incorporated the related description and citation to Lines 252-258.

Supplementary Fig. 5 The ROC curves of the Logistics Regression model classify CLR and DII patients using top 5,10,15,20 size-3 CC motifs as features, and CytoCommunity uses 29 dimensions. The Logistic Regression model uses features as motif counts from TrimNN and scales between 0 and 1.

3. In Figure 1B, the ROC curves give the initial impression that larger k-mer CC motifs lead to improved ROC values, suggesting that higher-order motifs provide stronger predictive power. However, in Figure 3a, this pattern does not seem to hold consistently. This raises the question of whether increasing motif complexity (higher k values) always enhances model performance, or if there are diminishing returns or trade-offs associated with using high-order CC motifs. The authors should provide a more detailed discussion on the advantages and disadvantages of high-order complex CC motifs, particularly in terms of predictive power, interpretability, and potential overfitting risks in smaller datasets. A deeper analysis of when and why higher-order motifs improve or fail to

improve performance would strengthen the biological and computational justification for TrimNN's motif-based approach.

Response: Thanks for your suggestion, which significantly improves the depth of our research.

In **Fig. 1B**, our ROC curve is only a schematic illustration, indicating that motifs of different sizes have distinct ROC curves. It does not imply that larger k-mer CC motifs lead to improved ROC values. To avoid misunderstanding, we have updated **Fig. 1B** by removing specific size values.

Fig. 1 TrimNN analysis workflow. **B**. These CC motifs can be biologically interpreted in the downstream analysis, including visualization, cellular level interpretation within cell-cell communication analysis, gene level interpretation within differentially expressed gene analysis, e.g., GO enrichment analysis and pathway enrichment analysis, and phenotypical analysis within the availability of phenotypical information, e.g., survival curve and phenotypic classification analysis. CC: cellular community.

Within these comprehensive experiments, along with three case studies, we discuss trade-offs between large and small sizes of CC motifs in presenting the characteristics of cellular neighborhoods in spatial biology.

(i) *Complex large-size CC motifs may bring better prediction power but are hard to identify.* In the case study of CRC, we confirmed that size-3 CC motifs have better AUROC than size-1 CC motifs to distinguish subtypes of CRC using the same number of features (**Fig. 3A**). For specific cell type combination, size-4 CC motifs can better distinguish survival than size-3 motifs, while size-2 cannot distinguish survival (**Fig. 3D, 3E, and 3F**). However, the number of motifs increases exponentially, which makes it more computationally expensive to identify the most overrepresented motifs in large sizes. For example, AD data used in the study has 13 cell types, which makes 13 size-1, 91 size-2, 455 size-3, 8,281 size-4, and 54,418 size-5 motifs.

(ii) *Complex large-size CC motifs have similar biological interpretability with smaller-size motifs but lack an established statistical framework in scientific rigor.* From the perspective of statistics, there are very limited approaches available to generalize statistical significance comparison across the combinatorial space in multiple sizes. We used Fisher's exact test (or the Chi-squared test) with odds ratios as the effect size within size-2 and size-3, and we are very hesitant and careful about further statistical evaluation in sizes larger than 3. Specifically, although a three-way contingency table can be analyzed using the Cochran–Mantel–Haenszel (CMH) test (PMID: 13655060) with the effect size measured by the Mantel–Haenszel common odds ratio, there is no single statistical test that is comparable to Fisher's exact test (or the Chi-squared test) or the CMH

test—that universally applies to four-way or higher-dimensional contingency tables. Furthermore, analyzing three-way or higher-order tables involves more complex hypothesis testing, as conditional and marginal associations may differ substantially (Agresti, Alan. Categorical data analysis. John Wiley & Sons, 2013.). However, we can use motif visualization, functional analysis in GO and pathway enrichment, cell-cell communication, and phenotypic analysis to easily interpret the motif results in multiple sizes. Notably, there are no limits in sizes to use these interpretation approaches on these motifs.

(iii) *Applying complex large-size CC motifs needs more caution in machine learning for overfitting.* Even CC motifs can be treated as representations of characteristics of cellular neighborhoods, users are supposed to be cautious about risks of overfitting when using them as features for machine learning tasks. When applying to current spatial omics datasets with a limited number of samples, the number of features as possible CC motifs in large size can often surpass the number of sample sizes. Take the same example of AD data with 13 cell types, it has 13 size-1, 91 size-2, 455 size-3, 8,281 size-4, and 54,418 size-5 motifs. Users need to be aware of selecting the appropriate number of top features to perform machine learning tasks to prevent overfitting.

The theory and practice of finding the CC motifs with the most appropriate size to describe the observed data in spatial biology is still in its infancy. As the size grows larger, the field currently lacks established approaches to perform comprehensive and rigorous computational, statistical, and biological evaluations. According to the reviewer’s guidance, we conducted a demo experiment using simulated graphs and motifs to explore the potential optimal size of CC motifs. (Supplementary Fig. 40).

Supplementary Fig. 40 Simulated triangulated graphs differ in cell type distribution in disease and control conditions. Specific size-3 to size-7 CC motifs are simulated to check their abundance in case and control conditions.

In this simulation data, we assumed that the disease and control target graphs have different distributions of cell types. We then counted the occurrences of our simulated specific size-3 to size-7 CC motifs in these two target triangulated graphs (**Supplementary Data 72**).

Supplementary Data 72. Occurrence numbers of size-3 and size-4 simulated CC motifs across disease and control samples in the simulation.

size	Disease	Control	Diff	Ratio (Disease+1/Control+1)
Size-3	18	9	9	1.90
Size-4	18	6	12	2.71
Size-5	18	3	15	4.75
Size-6	18	0	18	19.00
Size-7	3	0	3	4.00

As shown in the results, these CC motifs from size-3 to size-7 exhibit varying distinctions between disease and control samples, more than cell type compositions. From size-3 to size-6, the difference in occurrence numbers between the two conditions progressively increases with motif size. However, size-7 turned this trend of differences between disease and control. In general, the likelihood and frequency of motif occurrences in the target graph decrease when increasing the motif size, until a turning point is encountered to capture the largest numerical differences between the two conditions. In this case, we can select size-6, demonstrating the most significant differences between the case and the control.

Besides this specially designed system of size-7 to demo the idealized process in identifying the appropriate size of CC motifs, we also explored this question by expanding the identified size-3 motifs to subsets of size-12 in the context of AD research ranked by significance from functional enrichment (see our response to **Reviewer 1, Comment 1**), and extended the robustness simulation to size-5 to show the robustness of multi-size motifs resistant to noises in **Fig. 2A**, **Fig. 2B** and **Fig. 2C** (see our response to **Reviewer 1, Comment 2**).

In summary, we would like to emphasize that even large-size motifs reveal their values in top-down approaches, the theory and practice of finding the most appropriate size to describe the observed data in spatial biology is still in its infancy. From the limited experiments and observations, it may not have one optimal size of CC motifs that uniformly describes cellular communities within the complex biological context revealed from the spatial omics. Different approaches in diverse statistics, functional analysis, and phenotypic analysis could reveal different perspectives in different biological scenarios. We cannot give any specific recommendation with guarantees at this time, but we believe TrimNN provides an accurate and computationally feasible way to approach this challenge.

Theoretically, large-size CC motifs can be used in any spatial omics research with spatial coordinates and cell type annotations. Given all the experiments and case studies, we believe we have demonstrated the value of CC motifs in the proof-of-concept. CC motifs identified by TrimNN provide an expressive representation of the cellular community, which is robust, interpretable, and generalizable in spatial omics research.

We incorporated the related description into the paragraphs in the Discussion in Lines 654-682.

4. In Figure 4C and 4D, the spatial distribution of the ‘MMM’ CC motif shows a clear difference between the 13-month-old AD group and the control group, with microglia clustering more tightly in the AD group. However, in Supplementary Figure 8, there is no noticeable difference in the abundance of the MMM CC motif between the 8-month-old AD and control groups. This does not clearly verify whether the ‘MMM’ motif emerges in the early stages of AD. Moreover, the lack of a larger sample size for statistical analysis makes it difficult to ensure the stability of the ‘MMM’ motif across individual mice.

Response: Sorry for any misunderstanding caused by our lack of clear annotation. In **Supplementary Fig. 11D** (original Supplementary Fig. 8D), ‘MMM’ appears 34 times in the disease sample of the eight-month-old replicate 1 and 5 times in the control sample, with a Benjamini-Hochberg adjusted Fisher’s exact test p-value of $4.46e-05$ (**Supplementary Data 19**). In **Supplementary Fig. 11H** (original Supplementary Fig. 8H), ‘MMM’ appears 24 times in the disease sample of the eight-month-old replicate 2 and once in the control sample, with a Benjamini-Hochberg adjusted Fisher’s exact test p-value of $4.50e-06$ (**Supplementary Data 22**). This indicates that in the 8-month-old AD and control groups, ‘MMM’ still exhibits a significant difference, demonstrating that the ‘MMM’ CC motif may play an important role in both the early and late stages of AD. Unfortunately, until May 1st, 2025, we have not found additional publicly available AD datasets with different time points to further validate our results. But we are happy to include more relevant datasets, upon availability, to further demonstrate the value of our CC motifs in various disease models. However, based on the existing 8-month-old and 13-month-old replicate 1 and 2 data, we can observe the difference in the MMM CC motif between AD and control samples.

D

H
Supplementary Fig. 11. Representative CC motifs in triangulation graph in disease (left column) and control (right column) samples, including localization of motifs **D.** ‘MMM’ in eight-month-old sample replicate 1, **H.** ‘MMM’ in eight-month-old sample replicate 2.

5. In Figure 5, the authors identified two types of CC motifs: the Shifted Interaction Motif and the Homeostatic Interaction Motif, which were determined based on the ratio change between size-3 and size-4 motifs. Although the statistical results show significant trends, the study is based on only 2 CRC patients and 2 healthy controls, with a total of 58 ROIs, which may affect the generalizability of the conclusions. Additionally, there is no independent test set to validate these findings. The validation relied on statistical significance tests (proportion test, NCEM analysis), but no independent dataset was used for external validation.

Response: Thank you for the suggestion. To validate the generalizability of our conclusion, we explored an independent colorectal carcinoma study with their spatial proteomics data in CODEX, which is publicly available (PMID: 38036590). This study includes patient samples of normal and tumor regions at three tumor stages (CRC37_0092 at stage II, CRC35_0086 at stage III, CRC38_0028 at stage IV). In their provided CODEX dataset, T cells were not originally classified into CD4 and CD8 T cells. To validate our conclusion, we used Stellar (PMID: 36280720) to distinguish T cells into three subtypes: CD4 T cells, CD8 T cells, and other T cells. Within this independent dataset, we validated our conclusions on two types of CC motifs: the Shifted Interaction Motif and the Homeostatic Interaction Motif. The results of the two motifs are shown in **Supplementary Data 67** and **Supplementary Data 68**. The Homeostatic Interaction Motif remained consistent in abundance between the disease and control groups during the growth from size-3 to size-4, which is consistent with our observation. For the Shifted Interaction Motif, the conclusion from our manuscript is not as clearly reflected in this independent dataset. As CD4 T cells, CD8 T cells, and other immune cells were annotated by Stellar, this could introduce potential labeling errors. Additionally, since the samples in this dataset span three different tumor stages, the varying distributions across tumor stages may affect the establishment of a unified trend.

We incorporated the related description to Lines 579-583.

Supplementary Data 67. The occurrence of size-3 and size-4 Shifted Interaction Motifs across different samples in colorectal carcinoma. CRC37_0092 are samples from patients at stage II, CRC35_0086 are samples from patients at stage III, and CRC38_0028 are samples from patients at stage IV.

Sample	Size-3	Size-4
CRC36_86_reg001(normal)	8	12
CRC36_86_reg002(disease)	6	0
CRC37_92_reg001(normal)	0	0
CRC37_92_reg002(disease)	15	9
CRC38_28_reg001(normal)	4	4
CRC38_28_reg002(disease)	3	3
CRC38_28_reg003(disease)	0	0

Supplementary Data 68. Occurrences of size-3 and size-4 Homeostatic Interaction Motifs across different samples.

Sample	Size-3	Size-4
CRC36_86_reg001(normal)	8	6
CRC36_86_reg002(disease)	37	17
CRC37_92_reg001(normal)	1	1
CRC37_92_reg002(disease)	74	33
CRC38_28_reg001(normal)	14	7
CRC38_28_reg002(disease)	30	20
CRC38_28_reg003(disease)	38	17

Overall, TrimNN introduces a novel framework for identifying spatial motifs in omics data, demonstrating potential in biomarker discovery and disease characterization. However, its reliance on local topology may overlook long-range interactions, and the last two case studies are based on small datasets without independent validation, limiting generalizability.

Further validation on larger, independent datasets and methodological improvements to capture broader spatial dependencies would enhance its robustness and applicability. Given these limitations, a major revision is necessary to ensure the robustness and applicability of the proposed method.

Response: We appreciate the reviewer's insightful suggestions and we have made an effort to validate the robustness and applicability of our method. Please refer to our point-by-point responses above for details.

(Remarks on code availability)

I run the demo code and dataset, they work well

Response: We appreciate the reviewer's efforts in reviewing our code.

Reviewer #3:

Response: We sincerely appreciate the reviewer's constructive comments, which have been highly valuable in improving the quality of our manuscript.